# Mn$^{2+}$-activated dual-wavelength emitting materials toward wearable optical fibre temperature sensor

Enhai Song [1,5 ✉], Meihua Chen[1,5], Zitao Chen[1], Yayun Zhou[1], Weijie Zhou [1], Hong-Tao Sun [2], Xianfeng Yang [3], Jiulin Gan [1 ✉], Shi Ye [1] & Qinyuan Zhang [1,4 ✉]

Photothermal sensing is crucial for the creation of smart wearable devices. However, the discovery of luminescent materials with suitable dual-wavelength emissions is a great challenge for the construction of stable wearable optical fibre temperature sensors. Benefiting from the Mn$^{2+}$-Mn$^{2+}$ superexchange interactions, a dual-wavelength (530/650 nm)-emitting material Li$_2$ZnSiO$_4$:Mn$^{2+}$ is presented via simple increasing the Mn$^{2+}$ concentration, wherein the two emission bands have different temperature-dependent emission behaviours, but exhibit quite similar excitation spectra. Density functional theory calculations, coupled with extended X-ray absorption fine structure and electron-diffraction analyses reveal the origins of the two emission bands in this material. A wearable optical temperature sensor is fabricated by incorporating Li$_2$ZnSiO$_4$:Mn$^{2+}$ in stretchable elastomer-based optical fibres, which can provide thermal-sensitive emissions at dual- wavelengths for stable ratiometric temperature sensing with good precision and repeatability. More importantly, a wearable mask integrated with this stretchable fibre sensor is demonstrated for the detection of physiological thermal changes, showing great potential for use as a wearable health monitor. This study also provides a framework for creating transition-metal-activated luminescence materials.

[1] State Key Laboratory of Luminescent Material and Devices, and Guangdong Provincial Key Laboratory of Fibre Laser Materials and Applied Techniques, Guangdong Engineering Technology Research and Development Center of Special Optical Fiber Materials and Devices, South China University of Technology, 510641 Guangzhou, China. [2] International Center for Materials Nanoarchitectonics (MANA), National Institute for Materials Science (NIMS), 1-2-1 Sengen, Tsukuba 305-0047, Japan. [3] Analytical and Testing Centre, South China University of Technology, 510641 Guangzhou, Guangdong, China. [4] School of Physics and Optoelectronics, South China University of Technology, 510641 Guangzhou, China. [5] These authors contributed equally: Enhai Song, Meihua Chen. ✉email: msehsong@scut.edu.cn; msgan@scut.edu.cn; qyzhang@scut.edu.cn

D ual-wavelength emission materials have attracted increasing research interest because they can provide fluorescence intensity ratio (FIR) technology with self-calibration features for various applications[1–4]. In particular, FIR-technology-based temperature sensors have the distinctive advantages of not requiring contact, fast detection of moving objects, light weight, small size, immunity to electromagnetic interference, and resistance to harsh environments; these qualities make them preferable to conventional thermometers[5–8]. However, the discovery of suitable dual-wavelength emission materials for high-performance temperature sensors remains a challenge. Recently, dual-wavelength emission has been realised mainly by doping rare-earth ions ($Er^{3+}$ ($^4S_{3/2}/^2H_{11/2}$)[8–10], $Ho^{3+}$($^5G_6/^3K_8$)[11], $Nd^{3+}$ ($^4F_{7/2}/^4F_{3/2}$))[12] or transition-metal ions ($Cr^{3+}$($^2E/^4T_1$))[13,14] with two thermally coupled levels (TCL) into suitable host lattices, where electron populations at the lower and upper levels of TCL change inversely with increasing temperature, resulting in varied FIR values. The materials can be used to fabricate a temperature sensor with good absolute temperature sensitivity ($S_a$), but such sensor usually exhibits low relative temperature sensitivity ($S_r$)[15]. To address this issue, the strategy of doping two different emission species ($Eu^{2+}/Eu^{3+}$[16], $Pr^{3+}/Tb^{3+}$[17], $Mn^{4+}/Eu^{3+}$[18,19], $Ce^{3+}/Mn^{4+}$[20], etc.) into suitable compounds has been developed, because it has the potential to simultaneously achieve high sensitivity and good signal discriminability in temperature sensing. However, since the two emission bands of these materials possess different excitation spectra, their FIR values are affected by the wavelength of the excitation light source as well, thereby affecting the stability of the related temperature sensor. Therefore, design principle for suitable dual-wavelength emission materials is urgently required for stable temperature sensors.

As a photoluminescence activator, the transition-metal $Mn^{2+}$ is vital for modern lighting, displays, and imaging[21–29], owing to its unique visible luminescence properties in a solid matrix, such as tunable excitation and emission, high colour purity, and single-band emission. Additionally, as the $d$ electrons of $Mn^{2+}$ are not fully localised, an emission centre, the superexchange coupled $Mn^{2+}$-$Mn^{2+}$ dimer, might be formed when two $Mn^{2+}$ ions are close enough (~5 Å) and share anions[30]. However, this dimer usually shows a near-infrared emission band with thermal quenching behaviour that differs from the conventional visible emission from the isolated $Mn^{2+}$ ion[31,32]. Apparently, the dimer's emission is irrelevant for conventional lighting or display applications, but it is significant in many other fields. In principle, when $Mn^{2+}$ ions are unevenly distributed into a suitable structure, the emissions from the $Mn^{2+}$-$Mn^{2+}$ dimer with isolated $Mn^{2+}$ ions combine into dual-wavelength emissions, which is applicable in temperature sensors. However, as the distributions of $Mn^{2+}$ ions strongly depend on the crystal structure of the host lattice, the site-occupation of the dopant ($Mn^{2+}$) and its concentration, the design of $Mn^{2+}$-ion-based dual-wavelength emission materials remains a great challenge.

Herein, a dual-wavelength (530/650 nm) emitting material, $Li_2ZnSiO_4$:$Mn^{2+}$, is produced by controlling the doping concentration of $Mn^{2+}$, wherein the two emission bands have different concentration- and temperature-dependent luminescence behaviour, but exhibit quite similar excitation spectra. Although dual-emission characteristics have also been observed in two $Mn^{2+}$-doped crystallographic sites in some cases, the two emission bands always show distinct excitation spectra in these systems[33–36]. Additionally, rational control of the site preference of the dopant is difficult in these cases. We found that the emission ratio of the green and red emissions can be easily tuned by changing the $Mn^{2+}$ concentration in $Li_2ZnSiO_4$:$Mn^{2+}$. Experimental evidence from extended X-ray absorption fine structure (EXAFS), electron diffraction, and density functional

theory (DFT) calculations suggests that the green and red emissions in $Li_2ZnSiO_4$:$Mn^{2+}$ come from the isolated $Mn^{2+}$ ions and $Mn^{2+}$-$Mn^{2+}$ dimers, respectively. In particular, these two emission bands of $Li_2ZnSiO_4$:$Mn^{2+}$ have similar excitation spectra but feature different thermal quenching behaviours. By virtue of this photophysical characteristic, we fabricated a stable wearable optical fibre temperature sensor that shows good performance in both contact and noncontact temperature detection modes. This work not only provides new insights into the design of stable wearable optical fibre temperature sensors but also facilitates the luminescence control and application of $Mn^{2+}$-doped luminescence materials.

## Results

Figure 1a shows that the compound $Li_2ZnSiO_4$ (ICSD no. 8237) has a monoclinic structure with a $P1\ 21/n1(14)$ space group, and the lattice constants are $a = 6.262$ Å, $b = 10.602$ Å and $c = 5.021$ Å, $\beta = 90.51°$. It comprises $SiO_4$, $LiO_4$, $LiO_3$ and $ZnO_4$ units, which are connected by shared oxygen atoms. The site-occupancy preference of different concentrations of $Mn^{2+}$-doped $Li_2ZnSiO_4$ was simulated by DFT calculations. A $2 \times 2 \times 2$ supercell $Li_2ZnSiO_4$ ($Li_{64}Zn_{32}Si_{32}O_{128}$) was first constructed, and the possible substitution models for one or two cations replaced by one or two Mn ions were considered (Supplementary Figs. 1 and 2). Furthermore, because the ionic radius of $Li^+$ ($r = 0.59$ Å) is also similar to that of $Mn^{2+}$ ($r = 0.66$ Å), the substitution condition of $Li^+$ by $Mn^{2+}$ was considered in the simulation. In this substitution model, a $Li^+$ vacancy ($V_{Li}$) is introduced to maintain the charge balance. When the first $Mn^{2+}$ ion (Mn1) was introduced into the supercell $Li_{64}Zn_{32}SiO_{128}$, we calculated the formation energy ($E_f$) for the cases in which Mn occupied the Zn (M1), Li1 (M2, Mn/Li-$V_{Li}$) or Li2 (M3, Mn/Li2-$V_{Li}$) site in this structure. The $E_f$ value of substitution model M3 ($-0.89$ eV) is significantly smaller than that of M1 (6.56 eV) or M2 (6.80 eV) (see Fig. 1c), which strongly confirmed that at a low $Mn^{2+}$ concentration, $Mn^{2+}$ occupied the $Zn^{2+}$ site in this structure. Based on this result, we further calculated $E_f$ as the second Mn (Mn2) introduced in the substitution model M3 ($Li_8Zn_3MnSi_4O_{16}$). Five different possible substitution models were considered: $Mn/Zn^a$(M4, which had a relatively short Mn–Mn bond length), $Mn/Zn^b$(M5, which had a longer Mn–Mn bond length), Mn/Li(M6, Mn/Li-$V_{Li}$), Mn/Li2|Li2/Zn(M7, Mn occupied Li2 position and the replaced Li2 substituted Zn), and Mn/Li1|Li1/Zn (M8, Mn occupied the Li1 position and the replaced Li1 substituted Zn). The $E_f$ value of M7($-3.65$ eV) was much lower than that of the others, suggesting that the second $Mn^{2+}$ substituted Li2, and the substituted Li2 replaced $Zn^{2+}$ to maintain the charge balance. In this substitution model (M7), Mn1 and Mn2 have a distance of 1.87 Å and share one oxygen, which indicates that the $Mn^{2+}$(Zn)–$Mn^{2+}$(Li2) dimer can form at relatively high concentrations in $Mn^{2+}$-ions-doped samples. Supplementary Fig. 3 shows that different concentrations of $Mn^{2+}$ ion-doped $Li_2ZnSiO_4$ samples with a pure phase were obtained. Additionally, the Mn K-edge X-ray absorption near-edge spectroscopy (XANES) spectra of $Li_2Zn_{0.95}SiO_4$:$0.05Mn^{2+}$ and $Li_2Zn_{0.85}SiO_4$:$0.15Mn^{2+}$ are more similar to that of MnO than those of $Mn_2O_3$ or $MnO_2$ (see Fig. 1b). This observation indicates that the valence state of Mn in these samples is $+2$, which is consistent with the reduction-atmosphere synthesis conditions for these samples.

Figure 2a shows a high-resolution transmission electron microscopy (HR-TEM) image of a $Li_2Zn_{0.85}SiO_4$:$0.15Mn^{2+}$ particle. The interplanar distances of 0.30 and 0.23 nm are assigned to the (020) and (101) lattice planes of $Li_2ZnSiO_4$, respectively. The selected area electron-diffraction (SAED) patterns of the particle in the inset of Fig. 2a are given in Fig. 2c. The main

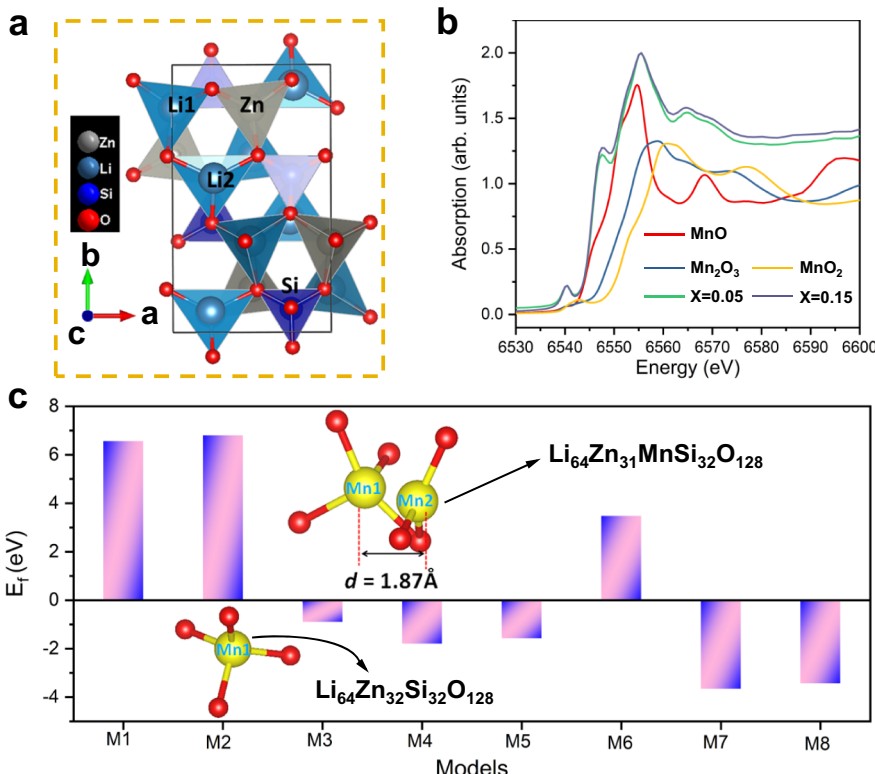

**Fig. 1 Structural analyses of Mn doping in Li₂ZnSiO₄. a** Crystal structure of Li₂ZnSiO₄. **b** The Mn K-edge XANES spectra of Li₂Zn$_{0.95}$SiO₄:0.05Mn²⁺ and Li₂Zn$_{0.85}$SiO₄:0.15Mn²⁺ and the reference compounds (MnO, Mn₂O₃ and MnO₂). **c** Formation energy ($E_f$) for different substitution models of Mn in Li₆₄Zn₃₂Si₃₂O₁₂₈ (M1–M3) and Li₆₄Zn₃₁MnSi₃₂O₁₂₈ (M4–M8).

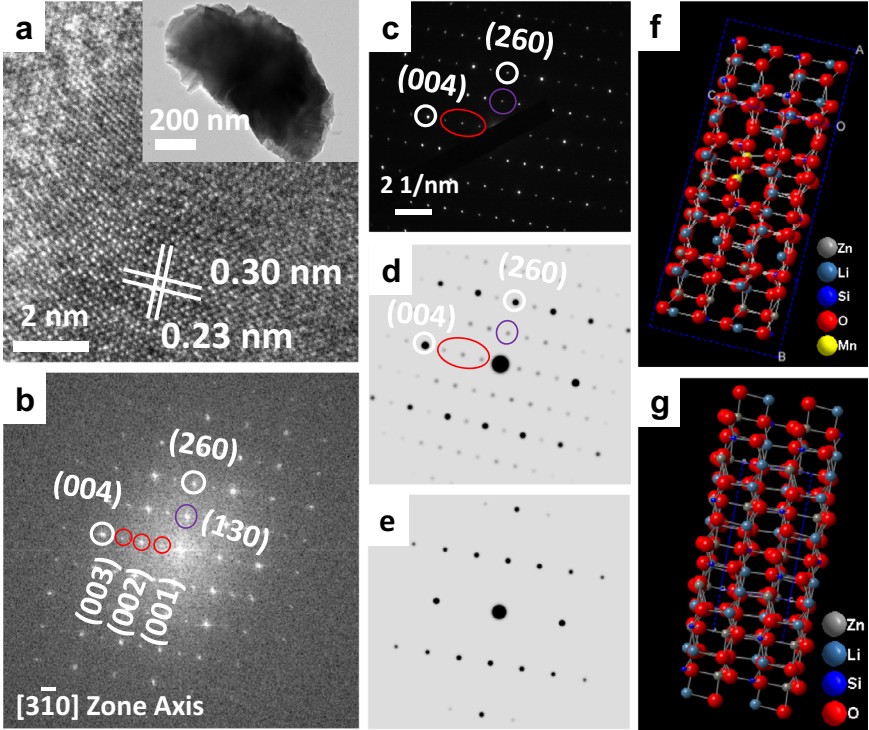

**Fig. 2 TEM characterisation of Li₂ZnSiO₄:Mn²⁺. a** HR-TEM image of a Li₂Zn$_{0.85}$SiO₄:0.15Mn²⁺ particle. The inset shows the low-resolution TEM image of the Li₂Zn$_{0.85}$SiO₄:0.15Mn²⁺ particle. **b** fast Fourier-transform diffraction patterns of the Li₂Zn$_{0.85}$SiO₄:0.15Mn²⁺ particle viewed along [3 1̄ 0] zone axis taken from the HR-TEM in Fig. 2a. **c** Selected area electron-diffraction pattern of the Li₂Zn$_{0.85}$SiO₄:0.15Mn²⁺. **d**, **f** The simulated electron-diffraction pattern and the corresponding crystallographic model of M7 viewed along [3 1̄ 0] zone axis. **e**, **g** The simulated electron-diffraction pattern and the corresponding crystallographic model of Li₂ZnSiO₄ viewed along [3 1̄ 0] zone axis.

diffraction maxima in the SAED patterns correspond to the diffraction patterns along the [3 $\bar{1}$ 0] zone axis of $Li_2ZnSiO_4$. However, one important additional feature observed in this sample is the presence of (001), (002), (003) and (130) superlattice spots (marked by the red and purple line circles). The appearance of these superlattice spots indicated a superstructure or defect. Figure 2b corresponds to the fast Fourier-transform (FFT) images of Fig. 2a, coinciding with the SAED in Fig. 2c. This evidence reveals that each individual particle exhibited a superstructure. The simulated electron-diffraction patterns (Fig. 2d, e) of model M7 (Fig. 2f) and the host lattice ($2 \times 2 \times 2$ supercell $Li_2ZnSiO_4$, Fig. 2g) used to identify the structure are provided. By comparison, the electron-diffraction patterns in Fig. 2b, c correspond well to the simulated patterns of M7 (Fig. 2d, f), indicating that the $Mn^{2+}$ ions substituted both Zn and Li2 sites in $Li_2Zn_{0.85}SiO_4:0.15Mn^{2+}$. Additionally, the electron-diffraction patterns showed that $Mn^{2+}$ ions would only occupy Zn sites in $Li_2Zn_{0.95}SiO_4:0.05Mn^{2+}$ (Supplementary Fig. 4). These findings are consistent with the EXAFS results (see Supplementary Fig. 5 and Supplementary Table 1) and DFT calculations, which further demonstrated that $Mn^{2+}$ ions occupied Zn sites in $Li_2Zn_{0.95}SiO_4:0.05Mn^{2+}$ and Zn and Li2 sites in $Li_2Zn_{0.85}SiO_4:0.15Mn^{2+}$. Therefore, single-to-dual wavelength emission can be expected in $Li_2ZnSiO_4:Mn^{2+}$ with increasing $Mn^{2+}$ ion concentration.

Figure 3a provides the concentration-dependent emission spectra of $Li_2Zn_{1-x}SiO_4:xMn^{2+}$ ($x = 0.01$–$0.30$) upon blue-light excitation at 427 nm. The sample with a relatively low $Mn^{2+}$ concentration ($x < 0.07$) comprises of a single green emission band centred at~530 nm with a full width at half maximum (FWHM) of ~36 nm, corresponding to the typical $^4T_1(^4G) \rightarrow {}^6A_1(^6S)$ transition of $Mn^{2+}$. With an increase in $Mn^{2+}$ concentration, the emission intensity increased initially and reached a maximum at $x = 0.07$. Afterwards, it gradually decreased because of concentration quenching. Therefore, the optimal concentration for green emission is $x = 0.07$. In addition to the green emission at ~530 nm, an additional red emission band at ~650 nm appeared in the emission spectra as the concentration of $Mn^{2+}$ was increased to $x \geq 0.07$. When the concentration of $Mn^{2+}$ is increased from $x = 0.07$ to $x = 0.30$, the emission intensity of the red emission increased gradually and reached a maximum at $x = 0.15$. Thereafter, it monotonically decreased. In addition, the emission peak position of the red emission shifted from 650 to 670 nm with increasing $Mn^{2+}$ concentration, due to the variations in the crystal field environment and the $Mn^{2+}$–$Mn^{2+}$ interactions[37]. Consequently, green-to-yellow then yellow-to-red emission were realised in this system by varying the $Mn^{2+}$ concentration (Supplementary Fig. 6). Additionally, the dual-wavelength emission of this structure can be realised over a wide range of doping concentrations, which may be beneficial for temperature sensors. The different concentration-dependent luminescence behaviours imply that the green and red emissions have different origins.

To obtain more information about the red emission, the excitation spectra of the two emission bands in $Li_2Zn_{1-x}SiO_4:xMn^{2+}$ ($x = 0.05$, 0.15, and 0.30) were collected and are shown in Fig. 3b. For the sample in which $x = 0.05$, the excitation spectrum of the green emission at ~528 nm consisted of five excitation bands at 357, 381, 427, 442 and 504 nm, corresponding to the $d$–$d$ transitions of $Mn^{2+}$ from the $^6A_1(^6S)$ ground state to the $^4E(^4D)$, $^4T_2(^4D)$, [$^4A_1(^4G)$, $^4E(^4G)$], $^4T_2(^4G)$ and $^4T_1(^4G)$ excited states, respectively. When $x = 0.15$, the green and red emissions of the sample possessed quite similar excitation spectra, suggesting that the absorptions were identical for the two emission bands. When the doping concentration was further increased to $x = 0.30$, no significant changes were observed in the excitation spectrum compared with the above samples, implying that both

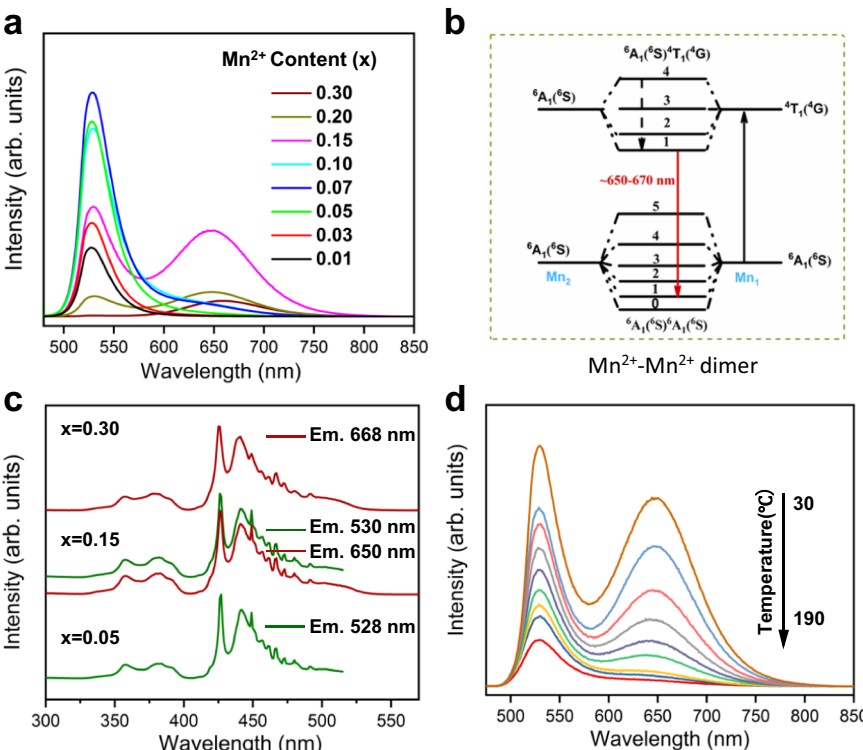

**Fig. 3 Photoluminescence properties and the mechanism of $Li_2ZnSiO_4:Mn^{2+}$. a** Emission spectra of $Li_2Zn_{1-x}SiO_4:xMn^{2+}$ ($x = 0.01$–$0.30$) and **b** excitation spectra of some typical $Li_2Zn_{1-x}SiO_4:xMn^{2+}$ ($x = 0.05$, 0.15, 0.30) samples. **c** Schematic diagram of $Mn^{2+}$-$Mn^{2+}$ dimer. **d** Temperature-dependent emission spectra of $Li_2Zn_{0.85}SiO_4:0.15Mn^{2+}$ upon 427 nm blue-light excitation.

emission bands originated from the $Mn^{2+}$ species. It is anomalous that the green and red emissions exhibit different luminescence decay behaviours (see Supplementary Fig. 7) but show similar excitation spectra. Additionally, no significant differences in excitation spectra of the two emission bands can be found even at a high temperature (Supplementary Fig. 8). According to the crystal structure shown in Fig. 1a and the typical excitation spectra of tetrahedral $Mn^{2+}$, the green emission was assigned to the tetrahedral $Mn^{2+}$. However, the red emission could not be ascribed to the octahedral $Mn^{2+}$ because the cations in $Li_2ZnSiO_4$ are four or three coordinated.

The DFT calculations, EXAFS and electron-diffraction analyses firmly demonstrated that $Mn^{2+}$–$Mn^{2+}$ dimers formed in $Li_2ZnSiO_4$:$Mn^{2+}$ at relatively high $Mn^{2+}$ concentrations. However, only isolated $Mn^{2+}$ was present in the sample with relatively low $Mn^{2+}$ concentrations. These results are consistent with the concentration-dependent emission spectra (see Fig. 3a). Therefore, the red emission is ascribed to the emission of the $Mn^{2+}$–$Mn^{2+}$ dimer. Figure 3c shows a luminescence diagram of the $Mn^{2+}$–$Mn^{2+}$ dimer[30]. The ground state $(^6A_1(^6S)^6A_1(^6S))$ of the $Mn^{2+}$–$Mn^{2+}$ dimer is formed by the exchange coupling of the ground states $(^6A_1(^6S))$ of Mn1 and Mn2, while the emitting state $(^6A_1(^6S)^4T_1(^4G))$ results from the coupling of the first excited state of Mn1 $(^4T_1(^4G))$ and the ground state of Mn2 $(^6A_1(^6S))$. Under these conditions, the excitation energies for both isolated $Mn^{2+}$ and $Mn^{2+}$–$Mn^{2+}$ dimer originated from the absorption of Mn1; thus, the green and red emissions possess identical excitation spectra. However, the two emission bands exhibited different quenching behaviours with increasing temperature, as shown in Fig. 3d. Specifically, as the temperature increased, the red emission band from the $Mn^{2+}$–$Mn^{2+}$ dimer quenched much faster than the green emission band (from isolated $Mn^{2+}$). Furthermore, when the temperature was increased to 190 °C, the red emission band was nearly fully quenched, and a pure green emission band was obtained. Such different thermal quenching behaviours of the two emission bands are due to their different luminescence mechanisms. For the dimer species (related to red emission), temperature strongly influenced the degree of interaction between the activator and host, and between the two $Mn^{2+}$ ions. A relatively high temperature can directly increase the $Mn^{2+}$-$Mn^{2+}$ distance, which will reduce the number of effective $Mn^{2+}$-$Mn^{2+}$ dimers[32]. In contrast, only the interactions between the host and activator can be considered for the isolated $Mn^{2+}$ ions with increasing temperature. Consequently, the red emission from the dimers is less thermally stable than that of the green emission from the isolated $Mn^{2+}$ ions.

Because of the similar excitation spectra and different thermal quenching behaviours of the two emission bands, the promising application of $Li_2Zn_{0.85}SiO_4$:$0.15Mn^{2+}$ in wearable temperature sensors was investigated. To endow the sensor with good light guidance and sensing performance, transparent optical encapsulant (OE) and polydimethylsiloxane (PDMS) with a high refractive index (RI) difference were selected as the matrix materials to fabricate flexible optical fibre[38], and the designed fibre with a double-cladding and double-tail fibre structure was provided in Fig. 4a. In this fibre, the core was fabricated using the OE with a high refractive index ($n_1 = 1.53$), as it has good light guiding properties. The inner cladding was fabricated by mixing the OE with $Li_2Zn_{0.85}SiO_4$:$0.15Mn^{2+}$ powder consisting of irregular particles ~1–5.5 μm- in size (Supplementary Fig. 9), and the inner cladding acted as the temperature-sensing fluorescence response layer. The outer cladding was prepared using PDMS, which has a relatively low RI ($n_2 = 1.41$). A section of two 5 mm long silica fibres with cores of 200 and 400 μm was inserted and integrated at the end of the fabricated flexible optical fibre. The silica fibres were used to guide the excitation light and collect the

fluorescence. The fabrication procedure for the optical fibre is shown in Supplementary Fig. 10. To validate the optical properties of the fibre, a frequency-doubled YAG laser at 532 nm was used as a light source to couple into the optical fibre (without the fluorescence response layer) to measure the transmission loss of the fibre through a cutback method (Fig. 4b, c). Figure 4d shows that the diameters the core and cladding of the fibre are 640 and 760 μm, respectively. Compared to the OE core-only structure (0.64 dB cm$^{-1}$), the OE/PDMS core-cladding structure shows a lower attenuation coefficient of 0.30 dB cm$^{-1}$ in air (Fig. 4e). The optical fibre with the protection of a PDMS cladding layer has a better ability to restrain the light compared to the OE core-only fibre; therefore, it meets the requirements of optical sensing. As shown in Fig. 4f, g, when we coupled a light-emitting diode (LED) with a central ultraviolet (UV) light with a wavelength of 365 nm into the fibre through the silica optical fibre, yellow–green fluorescence from the $Li_2Zn_{0.85}SiO_4$:$0.15Mn^{2+}$ optical fibre was observed. Figure 4h shows that the optical fibre had a core/inner cladding/out cladding diameter of 620/1050/1170 μm, exhibiting a smooth and uniform morphology along the fibre axis. Moreover, the optical fibre possessed good mechanical flexibility and deformability, and it could be easily bent and knotted (see Fig. 4i), showing good potential for wearable electronics.

The sensing platform based on this fabricated flexible optical sensing fibre and the compact all-fibre devices are shown in Fig. 5a. When an LED device with a central wavelength of 365 nm UV light was coupled into this flexible sensor through a pigtailed silica optical fibre with a core diameter of 200 μm, yellow–green fluorescence from the optical fibre was observed (Fig. 4g). Afterwards, another pigtailed silica optical fibre with a core diameter of 400 μm was used to collect fluorescence emission, which was connected to a high-pass filter and fluorescence spectrometer. Finally, the collection fluorescence signal was analysed in a processing unit. To accomplish temperature sensing and calibration, the sensor was placed in a metal heater; thus, the temperature was controlled from −20 to 100 °C with a precision of 0.1 °C. The emission spectra of the sensor in the temperature range of −20 to 100 °C are shown in Fig. 5b. All emission spectra consist of two strong emission bands at 530 and 650 nm, corresponding to the emission of the $Li_2Zn_{0.85}SiO_4$:$0.15Mn^{2+}$ phosphor in the fibre. The green and red emissions monotonically decreased because of the increasing nonradiative transition processes, which were similar to those of the pure powder sample (see Fig. 3d). The emission bands at 520–540 nm and 640–660 nm in the spectra were chosen for integral calculation, and marked as $I_{530}$ and $I_{650}$, respectively. Furthermore, the intensity ratio of $I_{530}/I_{650}$ was used as a sensing parameter. As shown in Fig. 5c, the sensor was calibrated in the range from −20 to 100 °C (each data point was measured three times to take the average value), showing good parabolic linearity, and it is well fitted as follows:

$$I_{530}/I_{650} = 2.348 \times 10^{-5} T^2 + 4.985 \times 10^{-4} T + 0.328 \quad (1)$$

Thus, one can easily measure or monitor the temperature by using the fabricated fibre. The absolute sensitivity ($S_a$) and relative sensitivity ($S_r$) of the fibre were calculated by the following equations:

$$S_a = \frac{dFIR}{dT} = \frac{d(I_{530}/I_{650})}{dT} \quad (2)$$

$$S_r = \left| \frac{1}{FIR} \frac{dFIR}{dT} \right| = \left| \frac{1}{(I_{530}/I_{650})} \frac{d(I_{530}/I_{650})}{dT} \right| \quad (3)$$

According to the above equations and Fig. 5c, the maximal $S_a$ and $S_r$ values at 100 °C were calculated as 0.0052 °C$^{-1}$ and 0.848% °C$^{-1}$, respectively, as shown in Supplementary Fig. 11a. Considering that the sensor may be used in the wearable field, we

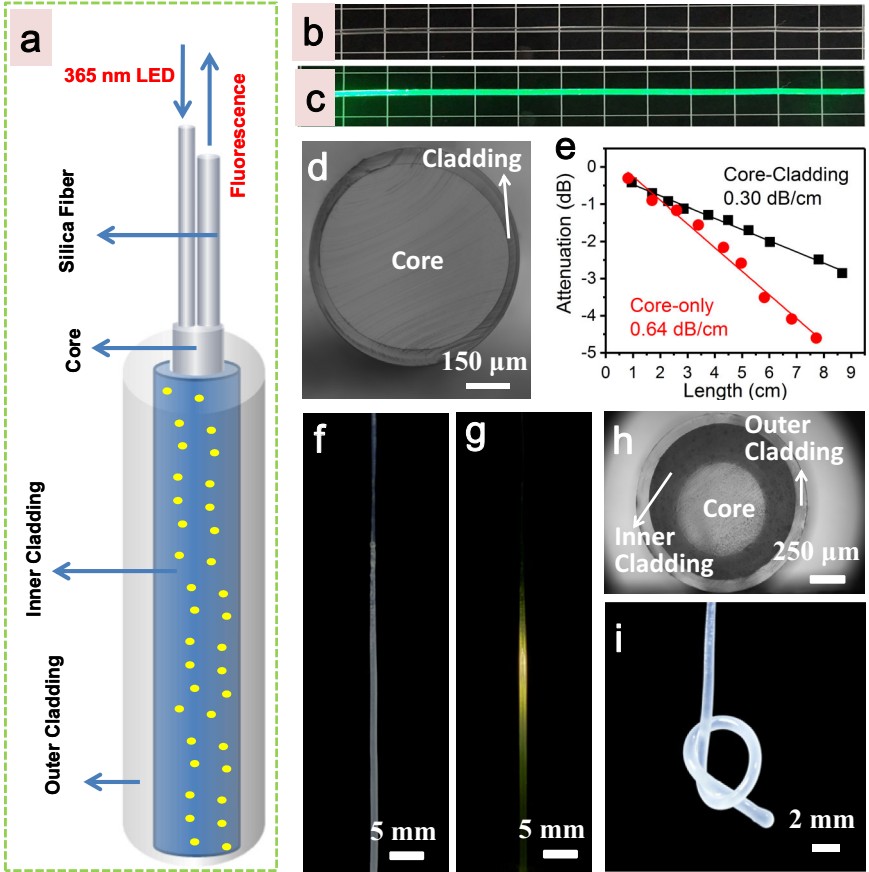

**Fig. 4 Flexible fibre structure based on Li₂ZnSiO₄:Mn²⁺.** **a** Structure of the designed flexible optical fibre. **b**, **c** Photographs of the OE fibre core with PDMS cladding and the light guiding in the optical fibre. Green laser was launched into the fibre through a coupled silica multimode optical fibre. **d** Photograph of the cross section of the OE fibre core with PDMS cladding. **e** Propagation loss of the OE fibre core with and without PDMS cladding measured by a cutback method in air condition. **f**, **g** Photographs of the as-fabricated fibre sensor with 365 nm LED off or on. **h** Cross section of the fabricated optical fibre sensor. **i** Photograph of the as-fabricated flexible optical fibre sensor integrated with Li₂Zn₀.₈₅SiO₄:0.15Mn²⁺ phosphor under natural-light irradiation.

further measured the $S_a$ and $S_r$ values of the sensor at approximately body temperature. As provided in Supplementary Fig. 11b, c, the $S_a$ and $S_r$ values within a small temperature range (34–44 °C) were calculated to be 0.0026 °C⁻¹ and 0.682% °C⁻¹, respectively; these values are comparable to those of previously reported electrical sensors[39] and rare-earth ion-doped silica optical fibre sensors[40]. Figure 5d shows the fluctuation of the sensor readout at a constant temperature of 37 °C, where the detection limit was approximately ±0.2 °C, as estimated from the standard deviation of signal drifting, suggesting that the temperature sensor has a good precision. Furthermore, the fluctuation is very small with increasing run-time, indicating that the fibre sensor is quite stable. Figure 5e exhibits the periodic response of the sensor to temperature. After five response cycles in the temperature range of 80 °C, the sensor maintained good reproducibility and showed stable and repeatable temperature detection properties. When the sensor was immersed in water at different temperatures to detect the water temperature in real time, it responded quickly and stably to various temperatures (see Fig. 5f). Moreover, as the sensor was placed near the human lip by fixing it to a mask, corresponding temperature changes of the exhaled and inhaled air were detected (Fig. 5g), indicating the potential application of this sensor for human health monitoring. The temperature response of the optical fibre sensor in the metal heater and the thermistor placed in situ were tested, as shown in Fig. 5h. The demodulation results showed that the flexible optical fibre sensor could mostly synchronise the changes with the

thermistor, and the optical fibre could respond accurately and quickly to the temperature. In addition, the fabricated fibre showed good biocompatibility and no cytotoxicity was observed (Supplementary Fig. 12), suggesting that the optical fibre can be safely used in wearable fields. Therefore, this flexible optical fibre integrated with the luminescent material Li₂Zn₀.₈₅SiO₄:0.15Mn²⁺ has promising applications as a wearable temperature sensor.

## Discussion

In summary, a series of Mn²⁺-doped Li₂ZnSiO₄ phosphors was synthesised by a facile solid-state reaction method. By varying the dopant content, single green emission at ~528 nm to dual emissions at 530 and 650 nm to pure-red emission at 670 nm was realised in this system. The two emission bands exhibited different luminescence decay and thermal quenching behaviours. However, they had quite similar excitation spectra. The static and dynamic luminescence investigations, crystal structure analysis, EXAFS, electron-diffraction measurements and DFT calculations indicated that the green and red emission bands can be ascribed to the isolated Mn²⁺ and exchange-coupled Mn²⁺-Mn²⁺ dimer, respectively.

Benefiting from the similar excitation spectra and different thermal quenching behaviours of the two emission bands, a stable flexible and wearable optical fibre temperature sensor with good precision (±0.2 °C) and repeatability as well as acceptable sensitivities was fabricated based on the luminescence material

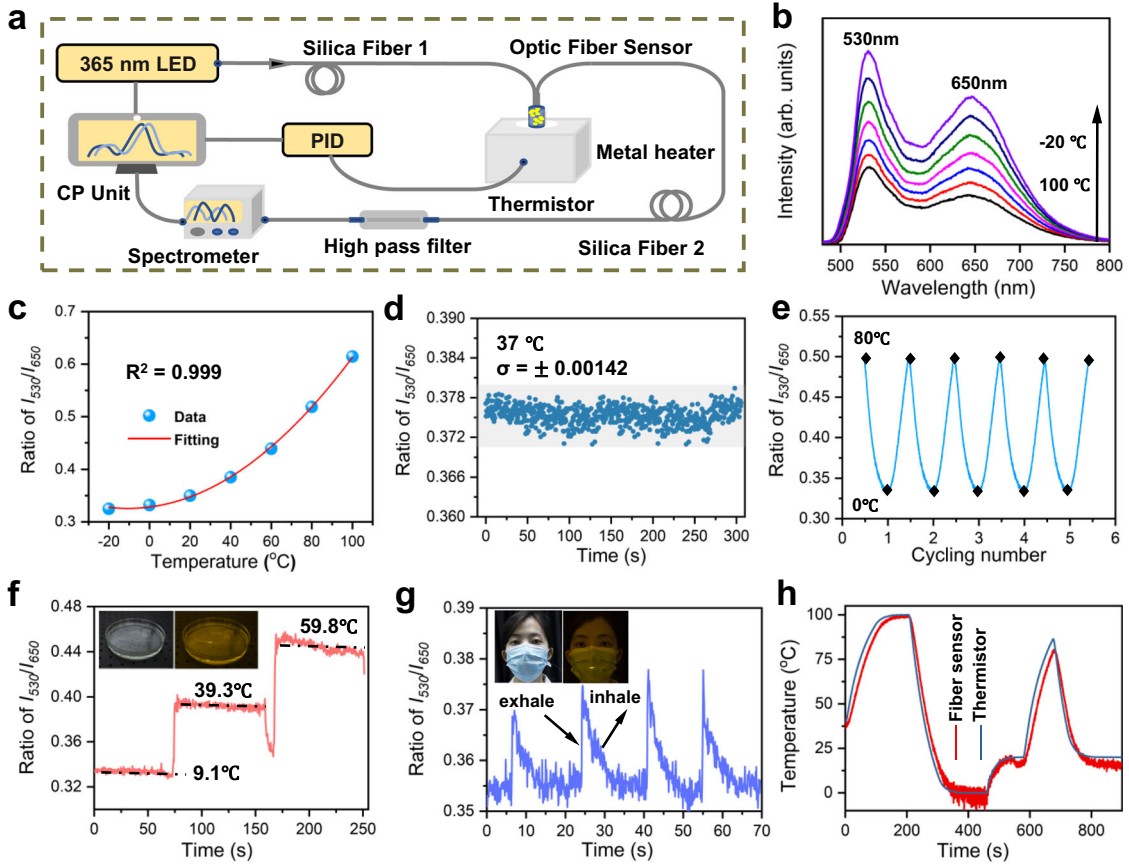

**Fig. 5 Performances of the wearable optical fibre temperature sensor. a** Schematic diagram of the optical fibre sensor based on fluorescence sensing platform. CP unit: Controlling and processing unit. PID: proportional-integral-derivative. **b** Fluorescence responses of the $Li_2Zn_{0.85}SiO_4$:$0.15Mn^{2+}$ integrated optical fibre sensor to temperature ranging from –20 to 100 °C with an interval of 20 °C. **c** Emission intensity ratios of $I_{530}/I_{650}$ versus the temperature. $I_{530}$ and $I_{650}$ at a given temperature are calculated based on the integrated emission intensity from 520–540 nm and 640–660 nm, respectively. **d** Fluctuations of the optical sensor output over time at the constant temperature of 37 °C. **e** Temperature cycling test of the optical fibre sensor between 0–80 °C. **f** Monitoring the change of water temperature with time by using the as-fabricated optical sensor. The inset shows the photograph of the water-temperature measuring by using the fibre sensor. **g** Monitoring the temperature change of the exhale and inhale with time by using the as-fabricated optical sensor. The inset displays the photograph of temperature change detection of the exhale and inhale of a volunteer by using the fibre sensor fixed on the mask. **h** Tracking the response time of a commercial thermistor and the fabricated optical fibre sensor.

$Li_2Zn_{0.85}SiO_4$:$0.15Mn^{2+}$ and some polymers. Additionally, the optical fibre temperature sensor showed good performance in real-time contact and non-contract temperature measurements, and it had good potential for monitoring human thermal activities. This work not only provides suitable materials for the fabrication of wearable temperature sensors but also presents different insights into the application and luminescence control of $Mn^{2+}$-doped luminescent materials.

## Methods

**Materials and preparation**. Phosphor samples $Li_2Zn_{1-x}SiO_4$:$xMn^{2+}$ ($x = 0.01$–0.30) are synthesised via a conventional solid-state reaction method and the raw materials are $Li_2CO_3$ (99.99% metals basis), $SiO_2$(AR, 99%), ZnO(99.9% metals basis) and $MnCO_3$(99.9% metals basis). All the raw materials were purchased from Aladdin Industrial Corporation (Shanghai, China) and used as received without further purification. The raw materials were weighted according to the nominal composition $Li_2Zn_{1-x}SiO_4$:$xMn^{2+}$, then ground and mixed well in a mortar. After that, the mixtures were put in a tube furnace to sinter at 1000 °C for 4 h under a reduction atmosphere ($H_2$:$N_2$ = 5:95). When the system was cooled down to the room temperature, the products were collected and reground for further characterisation.

**Characterisation**. The crystal structure and phase purity of the samples were characterised by a Rigaku D/max-IIIA X-ray diffractometer (XRD) with Cu-Kα radiation ($\lambda$ = 1.5418 Å). The morphology characterisations were measured by using scanning electron microscopy (SEM) (Nova, NANO SEM 430) and TEM

(JEOL, 2100F) methods. XAS measurements for the Mn K-edge were performed in fluorescence mode on beamline TLS 07A1 with electron energy of 1.5 GeV and an average current of 250mA, which is located in the National Synchrotron Radiation Research Centre (NSRRC) of Taiwan, China. The radiation was monochromatized by a Si (111) double-crystal monochromator. XANES and EXAFS data reduction and analysis were processed by Athena software. The photoluminescence excitation and emission, together with the luminescence decay curves were detected by a fluorescence spectrometer (FLS 920, Edinburgh Instruments). The luminescence thermal quenching behaviour of the sample is measured by the same spectrofluorimeter, which is equipped with a TAP-02 High-temperature fluorescence instrument (Tian Jin Orient–KOJI instrument Co., Ltd.).

**Fibre fabrication and characterisation**. For fibre fabrication, the following raw materials are used. Optical encapsulant (OE 6550 two-part silicone elastomer made of methylphenyl siloxane) and polydimethylsiloxane (PDMS Sylgard 184 two-part silicone elastomer) were obtained from Dow Corning Corporation (Shanghai, China). All the materials were used without any further purification. In the fabrication procedure, first of all, OE with 1:1 ratio of base and curing agent was configured as the core precursor, the precursor of the inner cladding is OE solution mixed with $Li_2Zn_{0.85}SiO_4$:$0.15Mn^{2+}$ at 10:1 quality ratio, and the outer cladding was PDMS with 10:1 quality ratio of base and curing agent. All the precursor solutions were mixed evenly by mechanical stirring for half an hour and degassed in vacuum. The Teflon tube is used as the mold to form the fibre core (100 °C, 2 h). After demoulding, the inner cladding (100 °C, 2 h) and the outer cladding (90 °C, 40 min) were spun and cured by heat. The emission spectra were obtained by a fibre optic spectrometer (QE Pro, Ocean Optics). The refractive indexes of PDMS and OE sheets were acquired employing a Metricon 2010/M prism coupler. The attenuation was measured with a Thorlabs PM100D optical power metre. The fabricated fibre was integrated onto mask to monitor the temperature changes of

human exhalation and inhalation. One volunteer participated in this study and was informed of the experiment details and asked to sign the consent.

**Computational methodology**. Theoretical calculations were performed by using density functional theory (DFT) implemented in the Vienna ab initio simulation package (Vasp)[41,42]. The exchange correlation potential was approximated by generalised gradient approximation (GGA) with the PBE functional[43]. The cutoff energy Ecut and k-point mesh were set to 400 eV and the $1 \times 1 \times 1$ Monkhorst-Pack grid, respectively, which are sufficient for energy convergence. The convergence criterion for the electronic energy was $10^{-5}$ eV and the structures were relaxed until the Hellmann–Feynman forces were smaller than 0.02 eV Å$^{-1}$. The formation energy ($E_f$) of the $Mn^{2+}$ doped $Li_2ZnSiO_4$ can be calculated by:

$$E_f = E(\text{doped}) - E(\text{perfect}) - \sum n_i \mu_i \tag{4}$$

where $E(\text{doped})$ and $E(\text{perfect})$ are the total energy of the doped and perfect (undoped) crystal, respectively. The $\mu_i$ and $n_i$ represent the chemical potential and the number of the added ($n_i > 0$) or removed ($n_i < 0$) $i$-type atoms, respectively.

## Data availability

The data generated and analysed during this study are available from the corresponding author upon reasonable request. The Source data underlying Figs. 1b, c, 3a, b, d, 4e, 5b–h are provided as a source data file. Source data are provided with this paper.

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

## Acknowledgements

This research was financially supported by the National Natural Science Foundation of China (Grant Nos. 51972117, and 52130201), Guangzhou science and technology planning project (202002030098), and State Key Lab of Luminescent Materials and Devices, South China University of Technology.

## Author contributions

E.S., J.G. and Q.Z. conceived and designed this project. E.S. directed with the experiments with contribution from M. C. built the optics fibre temperature sensor system. Z. C. and X. Y. performed the TEM and analysis; E.S., Y.Z., W.Z. and S.Y. contributed the luminescence mechanism analysis; E.S. performed the DFT calculations and interpretation of the computational data; E.S., H.S., J.G. and Q.Z. wrote the paper with contribution from all authors.

## Competing interests

The authors declare no competing interests.

## Additional information

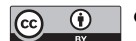

