## [Peer Review File · Nature Communications]

Reviewer #1 (Remarks to the Author):

Doping Concentration-Controlled Dual-Wavelength Emitting Materials toward Wearable Optical Fibre Temperature Sensor

Enhai Song, Meihua Chen, Zitao Chen, Yayun Zhou, Weijie Zhou, Hong-Tao Sun, Xianfeng Yang, Jiulin Gan, Shi Ye, Qinyuan Zhang

This manuscript contains a description of the conditions used for the preparation of $\text{Li}_2\text{ZnSiO}_4:\text{Mn}^{2+}$ phosphors by the solid state reaction method under reducing atmosphere ($\text{H}_2:\text{N}_2=5:95$). Several doping levels were tested. The structural properties of the powder products were evaluated by the XANES, EXAFS and TEM analyses. Several models of Mn^{2+} incorporation to the crystal lattice were evaluated by DFT methods. The photoluminescence parameters were measured by conventional experimental methods. The experimental conditions used for the sample preparation and property measurements are given appropriately that opens a possibility for future testing of the results by other researchers. In my opinion, this study is valuable because the two-band excitation-independent emission was observed in the phosphors prepared with different doping levels. The general level of this study is high and manuscript could be considered for publication after minor revision aimed at the increase of text quality. My several corrections proposed for the text are listed below for author consideration.

Page 2

Benefiting from the $\text{Mn}^{2+}-\text{Mn}^{2+}$ superexchange interactions, a dual-wavelength (530/650 nm)-emitting material, $\text{Li}_2\text{ZnSiO}_4:\text{Mn}^{2+}$, is demonstrated by simply increasing the Mn^{2+} concentration.

Benefiting from the $\text{Mn}^{2+}-\text{Mn}^{2+}$ superexchange interactions, a dual-wavelength (530/650 nm)-emitting material, $\text{Li}_2\text{ZnSiO}_4:\text{Mn}^{2+}$, is demonstrated by simple increasing the Mn^{2+} concentration.

Page 2

More importantly, wearable mask integrated with this stretchable fibre sensor is firstly demonstrated for the detection of physiological thermal changes, and it shows great potential for use as a wearable health monitor.

More importantly, wearable mask integrated with this stretchable fibre sensor is first time demonstrated for the detection of physiological thermal changes, and it shows great potential for use as a wearable health monitor.

Page 3

Dual-wavelength emission materials have gained increasing research interest because they can provide fluorescence intensity ratio (FIR) technology with self-calibration features for various

applications1-4.

Dual-wavelength emission materials have gained increasing research interest because they can provide fluorescence intensity ratio (FIR) technology with self-calibration features for various applications1-4.

Page 3

Recently, dual-wavelength emission has been mainly realised by doping rare-earth ions (Er^{3+} ($4\text{S}_3/2/2\text{H}_{11}/2$)⁸⁻¹⁰, Ho^{3+} ($5\text{G}_6/3\text{K}_8$)¹¹, Nd^{3+} ($4\text{F}_7/2/4\text{F}_3/2$)¹²) or transition metal ions (Cr^{3+} ($2\text{E}/4\text{T}_1$))^{13,14} with two thermally coupled levels (TCL) into suitable host lattices, where electron populations at the lower and upper levels of the TCL would change inversely with increasing temperature, resulting in varied FIR values.

Recently, dual-wavelength emission has been mainly realized by doping rare-earth ions (Er^{3+} ($4\text{S}_3/2/2\text{H}_{11}/2$)⁸⁻¹⁰, Ho^{3+} ($5\text{G}_6/3\text{K}_8$)¹¹, Nd^{3+} ($4\text{F}_7/2/4\text{F}_3/2$)¹²) or transition metal ions (Cr^{3+} ($2\text{E}/4\text{T}_1$))^{13,14} with two thermally coupled levels (TCL) into suitable host lattices, where electron populations at the lower and upper levels of TCL would change inversely with increasing temperature, resulting in varied FIR values.

Page 3

To address this issue, the strategy of doping two different emission species (like as $\text{Eu}^{2+}/\text{Eu}^{3+}$ ¹⁶, $\text{Pr}^{3+}/\text{Tb}^{3+}$ ¹⁷, $\text{Mn}^{4+}/\text{Eu}^{3+}$ ^{18,19}, $\text{Ce}^{3+}/\text{Mn}^{4+}$ ²⁰, etc.) into suitable compounds has been developed, because it has the potential to simultaneously achieve high sensitivity and excellent signal discriminability in temperature sensing.

To address this issue, the strategy of doping two different emission species (like as $\text{Eu}^{2+}/\text{Eu}^{3+}$ ¹⁶, $\text{Pr}^{3+}/\text{Tb}^{3+}$ ¹⁷, $\text{Mn}^{4+}/\text{Eu}^{3+}$ ^{18,19}, $\text{Ce}^{3+}/\text{Mn}^{4+}$ ²⁰, etc.) into suitable compounds has been developed, because it has the potential to simultaneously achieve high sensitivity and excellent signal discrimination in temperature sensing.

Page 4

Experimental evidence from extended X-ray absorption fine structure, and electron diffraction, as well as density functional theory (DFT) calculations, unambiguously suggests that the green and red emissions in $\text{Li}_2\text{ZnSiO}_4:\text{Mn}^{2+}$ come from the isolated Mn^{2+} ions and $\text{Mn}^{2+}-\text{Mn}^{2+}$ dimers, respectively.

Experimental evidence from extended X-ray absorption fine structure and electron diffraction, as well as density functional theory (DFT) calculations, unambiguously suggest that the green and red emissions in $\text{Li}_2\text{ZnSiO}_4:\text{Mn}^{2+}$ come from the isolated Mn^{2+} ions and $\text{Mn}^{2+}-\text{Mn}^{2+}$ dimers, respectively.

Page 5

The E_f of substitution model M3 (-0.89 eV) is significantly smaller than that of M1 (6.56 eV) or M2 (6.80 eV) (see Fig. 1b), which firmly confirmed that at a low Mn^{2+} concentration, Mn^{2+} occupied the

Zn²⁺ site in this structure.

The E_f value of substitution model M3 (-0.89 eV) is significantly smaller than that of M1 (6.56 eV) or M2 (6.80 eV) (see Fig. 1b), which firmly confirmed that at a low Mn²⁺ concentration, Mn²⁺ occupied the Zn²⁺ site in this structure.

Page 5

Based on this result, we further calculated the E_f as the second Mn (Mn₂) was introduced in substitution model M3 (Li₈Zn₃MnSi₄O₁₆).

Based on this result, we further calculated E_f as the second Mn (Mn₂) was introduced in the substitution model M3 (Li₈Zn₃MnSi₄O₁₆).

Page 5

The E_f of M7(-3.65 eV) is much lower than that of others, suggesting that the second Mn²⁺ substituted Li₂, and the substituted Li₂ replaced Zn²⁺ to maintain the charge balance.

The E_f value of M7(-3.65 eV) is much lower than that of others, suggesting that the second Mn²⁺ substituted Li₂, and the substituted Li₂ replaced Zn²⁺ to maintain the charge balance.

Page 6

Under this substitution model (M7), Mn₁ and Mn₂ have a distance of 1.87 Å and share with one oxygen, which suggests that Mn²⁺(Zn)–Mn²⁺(Li₂) dimer can be formed in relatively high-concentration in Mn²⁺–ions–doped samples.

Under this substitution model (M7), Mn₁ and Mn₂ have a distance of 1.87 Å and share with one oxygen, which suggests that Mn²⁺(Zn)–Mn²⁺(Li₂) dimer can be formed at relatively high-concentrations in Mn²⁺–ions–doped samples.

Page 6

The XANES spectra of Li₂Zn_{0.95}SiO₄:0.05Mn²⁺ and Li₂Zn_{0.85}SiO₄:0.15Mn²⁺ are considerably similar to that of MnO than those of Mn₂O₃ or MnO.

The XANES spectra of Li₂Zn_{0.95}SiO₄:0.05Mn²⁺ and Li₂Zn_{0.85}SiO₄:0.15Mn²⁺ are considerably similar to that of MnO than those of Mn₂O₃ or MnO₂.

Page 6

The detailed fitting parameters are summarised Supplementary Table 1.

The detailed fitting parameters are summarized in Supplementary Table 1.

Page 7

Fig. 2a shows a high-resolution transmission electron microscope (HR-TEM) image of a $\text{Li}_2\text{Zn}_{0.95}\text{SiO}_4:0.05\text{Mn}^{2+}$ particle, in which the interplanar distances are determined as 0.52 and 0.38 nm, corresponding to the (020) and (101) lattice planes of $\text{Li}_2\text{ZnSiO}_4$, respectively.

Fig. 2a shows a high-resolution transmission electron microscopy (HR-TEM) image of an $\text{Li}_2\text{Zn}_{0.95}\text{SiO}_4:0.05\text{Mn}^{2+}$ particle, in which the interplanar distances are determined as 0.52 and 0.38 nm, corresponding to the (020) and (101) lattice planes of $\text{Li}_2\text{ZnSiO}_4$, respectively.

Page 7

Figs. 2g and h further show the HR-TEM images of a $\text{Li}_2\text{Zn}_{0.85}\text{SiO}_4:0.15\text{Mn}^{2+}$ particle.

Figs. 2g and 2h further show the HR-TEM images of a $\text{Li}_2\text{Zn}_{0.85}\text{SiO}_4:0.15\text{Mn}^{2+}$ particle.

Page 7

Specifically, some additional weak points (highlighted with red and purple line circles) can be observed in Figs. 2i and j.

Specifically, some additional weak points (highlighted with red and purple line circles) can be observed in Figs. 2i and 2j.

Page 9

When the doping concentration is further increased to $x = 0.30$, no significant change is found in the excitation spectrum compared with the above samples, implying that both emission bands originated from the Mn^{2+} species.

When the doping concentration is further increased to $x = 0.30$, no significant change is found in the excitation spectrum compared with the above samples, implying that both emission bands originated from the Mn^{2+} species.

Page 9

The luminescence decay curves of the two emission bands are provided in Figs. 3c and d, respectively.

The luminescence decay curves of the two emission bands are provided in Figs. 3c and 3d, respectively.

Page 10

The monotonous decrease in the emission intensities of the two emission bands with increasing temperature is mainly ascribed to the gradually increasing nonradiative processes.

The monotonous decrease in the emission intensities of the two emission bands with increasing temperature is mainly ascribed to the gradually increasing nonradiative processes.

Page 10

A relatively high temperature can directly increase the Mn²⁺-Mn²⁺ distance, which will reduce the number of effective Mn²⁺-Mn²⁺ dimers³².

A relatively high temperature can directly increase the Mn²⁺-Mn²⁺ distance, which will reduce the number of effective Mn²⁺-Mn²⁺ dimers³².

Page 10

Consequently, the red emission from the dimers has worse thermal stability than that the green one (from the isolated Mn²⁺ ions).

Consequently, the red emission from the dimers has worse thermal stability than that of the green one (from the isolated Mn²⁺ ions).

Page 11

Because of the similar excitation spectra and different thermal quenching behaviours of the two emission bands, the promising application of Li₂Zn_{0.85}SiO₄:0.15Mn²⁺ in wearable temperature sensors has been investigated and demonstrated.

Because of the similar excitation spectra and different thermal quenching behaviors of the two emission bands, the promising application of Li₂Zn_{0.85}SiO₄:0.15Mn²⁺ in wearable temperature sensors was investigated.

Page 11

To acquire good light guiding and sensing performance of the sensor, a transparent optical encapsulant (OE) and polydimethylsiloxane (PDMS) with a high refractive index (RI) difference have been selected as the matrix materials to fabricate flexible optical fibre³⁸, and the designed fibre with a double-cladding and double-tail fibre structure is provided in Fig. 4a.

To acquire good light guiding and sensing performance of the sensor, a transparent optical encapsulant (OE) and polydimethylsiloxane (PDMS) with a high refractive index (RI) difference were selected as the matrix materials to fabricate flexible optical fibre³⁸, and the designed fibre with a double-cladding and double-tail fibre structure is provided in Fig. 4a.

Page 11

In this fibre, the core was fabricated using an OE with a high refractive index ($n_1 = 1.53$), which has excellent light guiding properties. The inner cladding was fabricated by mixing the OE with Li₂Zn_{0.85}SiO₄:0.15Mn²⁺ powder consisting of irregular particles with a size of ~1–5.5 μm (Supplementary Fig. 5), and the inner cladding acted as the temperature-sensing fluorescence response layer.

In this fibre, the core was fabricated using OE with a high refractive index ($n_1 = 1.53$), which has

excellent light guiding properties. The inner cladding was fabricated by mixing OE with $\text{Li}_2\text{Zn}_{0.85}\text{SiO}_4:0.15\text{Mn}^{2+}$ powder consisting of irregular particles with a size of $\sim 1\text{--}5.5\ \mu\text{m}$ (Supplementary Fig. 5), and the inner cladding acted as the temperature-sensing fluorescence response layer.

Page 11

The silica fibres were used to guide the excitation light and collect fluorescence.

The silica fibres were used to guide the excitation light and collect fluorescence.

Page 12

The numerical aperture ($\text{NA} = \sqrt{n_1^2 - n_2^2}$) of the designed optical fibre is about 0.59, exhibiting a good ability to collect and guide light.

The numerical aperture ($\text{NA} = \sqrt{n_1^2 - n_2^2}$) of the designed optical fibre is about 0.59, exhibiting a good ability to collect and guide light.

Page 12

When an LED with a central wavelength of 365 nm UV-light was coupled into this flexible sensor through pigtailed silica optical fibre with a core diameter of 200 μm , yellow-green fluorescence from the optical fibre can be observed, corresponding to the emission of $\text{Li}_2\text{Zn}_{0.85}\text{SiO}_4:0.15\text{Mn}^{2+}$ (Fig. 4h).

When an LED device with a central wavelength of 365 nm UV-light was coupled into this flexible sensor through pigtailed silica optical fibre with a core diameter of 200 μm , yellow-green fluorescence from the optical fibre can be observed, corresponding to the emission of $\text{Li}_2\text{Zn}_{0.85}\text{SiO}_4:0.15\text{Mn}^{2+}$ (Fig. 4h).

Page 13

As shown in Fig. 5c, the sensor is calibrated in the range of -20 to $100\ ^\circ\text{C}$ (each data point was measured three times to take the average value), showing good parabola linearity, and it is well-fitted:

$$524530650I/I = 2.3483 \cdot 10^{-4} T + 4.9851 \cdot 10^{-4} T + 0.3282 .$$

As shown in Fig. 5c, the sensor is calibrated in the range of -20 to $100\ ^\circ\text{C}$ (each data point was measured three times to take the average value), showing good parabola linearity, and it is well-fitted by relation: $524530650I/I = 2.3483 \cdot 10^{-4} T + 4.9851 \cdot 10^{-4} T + 0.3282 .$

Page 13

The temperature response of the optical fibre sensor in the metal heater and the in-situ placed thermistor have been tested as shown in Fig. 5g.

The temperature response of the optical fibre sensor in the metal heater and the in-situ placed thermistor have been tested, as shown in Fig. 5g.

Page 14

In summary, a series of Mn²⁺ doped Li₂ZnSiO₄ phosphors have been synthesized by a facile solid-state reaction method.

In summary, a series of Mn²⁺ doped Li₂ZnSiO₄ phosphors was synthesized by a facile solid-state reaction method.

Page 14

Benefiting from the similar excitation spectra and different thermal quenching behaviours of the two emission bands, a stable flexible and wearable optical fibre temperature sensor was fabricated based on the luminescence material Li₂Zn_{0.85}SiO₄:0.15Mn²⁺ and some polymers.

Benefiting from the similar excitation spectra and different thermal quenching behaviors of the two emission bands, a stable flexible and wearable optical fibre temperature sensor was fabricated based on the luminescence material Li₂Zn_{0.85}SiO₄:0.15Mn²⁺ and some polymers.

Page 14

Additionally, the optical fibre temperature sensor shows good performances in real-time contact and non-contact temperature measurements, and which has good potential for monitoring of human thermal activities.

Additionally, the optical fibre temperature sensor shows good performances in real-time contact and non-contact temperature measurements, and it has good potential for monitoring of human thermal activities.

Page 15

Phosphor samples Li₂Zn_{1-x}SiO₄:xMn²⁺ (x = 0.01-0.30) are synthesized via a conventional solid state reaction method and the raw materials are Li₂CO₃ (99.99%), SiO₂(AR), ZnO(99.9%) and MnCO₃(99.9%), and all the raw materials are used as received without further purification.

Phosphor samples Li₂Zn_{1-x}SiO₄:xMn²⁺ (x = 0.01-0.30) are synthesized via a conventional solid state reaction method and the raw materials are Li₂CO₃ (99.99%), SiO₂(AR), ZnO(99.9%) and MnCO₃(99.9%), and all the raw materials are used as received without further purification.

Besides purity, supplier should be reported for each raw material.

Page 15

The raw materials are weighted according to the nominal composition Li₂Zn_{1-x}SiO₄:xMn²⁺, then ground and mix it well in a mortar.

The raw materials are weighted according to the nominal composition Li₂Zn_{1-x}SiO₄:xMn²⁺, then ground and mixed well in a mortar.

Page 15

The morphology characterizations were measured by using the SEM (Nova, NANO SEM 430) and TEM (JEOL, 2100F).

The morphology characterizations were measured by using SEM (Nova, NANO SEM 430) and TEM (JEOL, 2100F) methods.

Page 15

The luminescence thermal quenching behavior and photoluminescence quantum yield (QY) of the sample are measured by the same spectrofluorimeter which are equipped with a TAP-02 High-temperature fluorescence instrument (Tian Jin Orient – KOJI instrument Co., Ltd.) and an integrated sphere, respectively.

The luminescence thermal quenching behavior and photoluminescence quantum yield (QY) of the sample are measured by the same spectrofluorimeter which is equipped with a TAP-02 High-temperature fluorescence instrument (Tian Jin Orient – KOJI instrument Co., Ltd.) and an integrated sphere.

Reviewer #2 (Remarks to the Author):

The Authors report possibility to use fluorescence intensity ratio (FIR) material to develop wearable optical temperature sensor. FIR materials are hot topic with big amount of publications and interesting possibilities.

The Authors give theoretical background and details about FIR material LiZnSiO:Mn, but the main result is usage of LiZnSiO:Mn for wearable health monitoring. The Authors prepare flexible optical fiber with LiZnSiO:Mn in inner core of the fiber. Specially during time of Covid-19, Fig. 5f is extremely interesting.

The problem in the present manuscript is that the main part, developing of optical fiber for wearable temperature sensing, is only small part of the text. Main part is analysis of LiZnSiO:Mn. To what extend SEM, TEM, XANES, EXAFS, DFT and Vasp are crucial for the main result? Or should part of them be separated to separate manuscript or to Supplementary Information? These results are not mentioned in Abstract. The present form of the manuscript is rather long. If the Authors analyze LiZnSiO:Mn, then they should also analyze which material would be most suitable for optical fiber. Why LiZnSiO:Mn? At present time exist rather big selection of FIR materials. Usually in FIR analyses is give value for absolute and relative temperature sensitivity, S_a and S_r . Now the Authors give only reproducibility (Fig. 5d).

Small problem in the manuscript is that it is not very carefully written. On line 49 is written because, line 64 to simultaneously, on line 136 should read MnO₂ and on line 172 should read 2i and k, on lines 183 and 188 is green emission at 525 and 530 nm (which is correct?), on line 205 a line should be replaced by a comma, on line 231 word lower should be removed, on line 244 is word intensities, on line 245 is word mainly, on line 265 is word caldding, line 320 and Fig. 5c contains too many decimals, on line 348 is word optical, on line 361 is word viay, on line 407 mjuu is missing, and on

line 444 concentration 005 should be replaced by 015. Most of the figures have too small fonts. In Fig. 2b scale bar is not clearly visible, Figs 2g and 2h looks similar, Em 530 nm (green line) in Fig. 3b is not visible.

Finally, the title could be more compact. At least, three first words (Doping Concentration-Controlled) are not needed, because main result is not the concentration analysis.

The manuscript is interesting and it will attract big amount of attention, but it should be written more carefully and in more compact form.

Reviewer #3 (Remarks to the Author):

The author reported a novel dual-wavelength emitting material $\text{Li}_2\text{ZnSiO}_4:\text{Mn}^{2+}$ as wearable optical fiber temperature sensors. It is interesting that the two emission bands (525/650 nm) of the material have different concentration quenching and temperature quenching behaviors, but exhibit similar excitation spectra. Based on DFT calculations, EXAFS and electron diffraction measurements, the author clearly demonstrated that the green and red emissions were ascribed to the isolated Mn^{2+} and $\text{Mn}^{2+}-\text{Mn}^{2+}$ dimer, respectively. It is crucial for the understanding and design of the luminescence behavior and the luminescence tuning of transition metal Mn^{2+} activated phosphors. Moreover, based on the new fiber fabrication procedure, the author extended the application of Mn^{2+} doped phosphors to the wearable optical fiber temperature sensor, showing high promise in human health monitoring. Overall, this is a well-prepared manuscript with an innovative demonstration for me. This paper is unique and may be publishable with minor revisions. My concerns are summarized in the following.

- 1) The corresponding lattice planes of the extra electron diffraction points in Figure 2k, i should be identified and marked.
- 2) The introductions of the insets in Figure 5e, f should be added in the corresponding Figure caption part.
- 3) As shown in Figure 1a, the crystal structure of $\text{Li}_2\text{ZnSiO}_4$ is provided, while no reference or card number (ICSD? or others) about this structure is shown in the manuscript. The information about the source of the crystal structure should be given.
- 4) Figure 3b, only relatively weak excitation at ~ 365 nm of the phosphor $\text{Li}_2\text{ZnSiO}_4:\text{Mn}^{2+}$ is shown, but why the authors used the 365 nm UV-LED as the excitation source in the fabricated wearable optical fiber temperature sensor?
- 5) As shown in Figure 3b, the two emission bands of $\text{Li}_2\text{ZnSiO}_4:\text{Mn}^{2+}$ have quite similar excitation spectra at room temperature. Are there some differences in the excitation spectra of the two emission bands at relatively high temperature (such as 100°C)?
- 6) The authors should explain more clearly the reason why the emission bands of $\text{Mn}^{2+}-\text{Mn}^{2+}$ dimer and isolated Mn^{2+} have quite similar excitation spectra in this system.
- 7) The authors addressed that Mn^{2+} could occupy both the Li and Zn sites for samples with high concentration of Mn^{2+} . Are there any differences in PL spectra of these two sites?
- 8) For the dimer's emission, does it exist in all Mn^{2+} doped phosphors as the doping level of Mn^{2+} is high enough?
- 9) Please comment on the safety of the proposed optical temperature sensor used in human health monitoring field.

Reviewer #1 (Remarks to the Author):

334958_0_art_file_5962455_r0r96r

Doping Concentration-Controlled Dual-Wavelength Emitting Materials toward Wearable Optical Fibre Temperature Sensor

Enhai Song, Meihua Chen, Zitao Chen, Yayun Zhou, Weijie Zhou, Hong-Tao Sun, Xianfeng Yang, Jiulin Gan, Shi Ye, Qinyuan Zhang

This manuscript contains a description of the conditions used for the preparation of $\text{Li}_2\text{ZnSiO}_4:\text{Mn}^{2+}$ phosphors by the solid state reaction method under reducing atmosphere ($\text{H}_2:\text{N}_2=5:95$). Several doping levels were tested. The structural properties of the powder products were evaluated by the XANES, EXAFS and TEM analyses. Several models of Mn^{2+} incorporation to the crystal lattice were evaluated by DFT methods. The photoluminescence parameters were measured by conventional experimental methods. The experimental conditions used for the sample preparation and property measurements are given appropriately that opens a possibility for future testing of the results by other researchers. In my opinion, this study is valuable because the two-band excitation-independent emission was observed in the phosphors prepared with different doping levels. The general level of this study is high and manuscript could be considered for publication after minor revision aimed at the increase of text quality. My several corrections proposed for the text are listed below for author consideration.

Author Reply: Thanks for your valuable comments and suggestions. Some necessary modifications have been made according to your suggestion and highlighted in blue color.

1. Page 2:

Original text: Benefiting from the Mn^{2+} - Mn^{2+} superexchange interactions, a dual-wavelength (530/650 nm)-emitting material, $\text{Li}_2\text{ZnSiO}_4:\text{Mn}^{2+}$, is demonstrated by simply increasing the Mn^{2+} concentration.

Suggested modification: Benefiting from the Mn^{2+} - Mn^{2+} superexchange interactions, a dual-wavelength (530/650 nm)-emitting material, $\text{Li}_2\text{ZnSiO}_4:\text{Mn}^{2+}$, is demonstrated by simple increasing the Mn^{2+} concentration.

Author Reply: Thanks. On page 2, the word “simply” has been corrected as “simple”.

2. Page 2

Original text: More importantly, wearable mask integrated with this stretchable fibre sensor is firstly demonstrated for the detection of physiological thermal changes, and it shows great potential for use as a wearable health monitor.

Suggested modification: More importantly, wearable mask integrated with this stretchable fibre sensor is first time demonstrated for the detection of physiological thermal changes, and it shows great potential for use as a wearable health monitor.

Author Reply: Thanks. On page 2, the “firstly” has been deleted in the revised manuscript.

3. Page 3

Original text: Dual-wavelength emission materials have gained increasing research interest because they can provide fluorescence intensity ratio (FIR) technology with self-calibration features for various applications1-4.

Suggested modification: Dual-wavelength emission materials have gained increasing research interest because they can provide fluorescence intensity ratio (FIR) technology with self-calibration features for various applications1-4.

Author Reply: On page 3, the word “because” has been corrected as “**because**”.

4. Page 3

Original text: Recently, dual-wavelength emission has been mainly realised by doping rare-earth ions ($\text{Er}^{3+}({}^4\text{S}_{3/2}/{}^2\text{H}_{11/2})$ 8-10, $\text{Ho}^{3+}({}^5\text{G}_6/{}^3\text{K}_8)$ 11, $\text{Nd}^{3+}({}^4\text{F}_{7/2}/{}^4\text{F}_{3/2})$ 12 or transition metal ions ($\text{Cr}^{3+}({}^2\text{E}^4/\text{T}_1)$)13,14 with two thermally coupled levels (TCL) into suitable host lattices, where electron populations at the lower and upper levels of the TCL would change inversely with increasing temperature, resulting in varied FIR values.

Suggested modification: Recently, dual-wavelength emission has been mainly realized by doping rare-earth ions ($\text{Er}^{3+}({}^4\text{S}_{3/2}/{}^2\text{H}_{11/2})$ 8-10, $\text{Ho}^{3+}({}^5\text{G}_6/{}^3\text{K}_8)$ 11, $\text{Nd}^{3+}({}^4\text{F}_{7/2}/{}^4\text{F}_{3/2})$ 12 or transition metal ions ($\text{Cr}^{3+}({}^2\text{E}^4/\text{T}_1)$)13,14 with two thermally coupled levels (TCL) into suitable host lattices, where electron populations at the lower and upper levels of TCL would change inversely with increasing temperature, resulting in varied FIR values.

Author Reply: On page 3, the word “realised” has been changed to “**realized**”. In the sentence “...upper levels of the TCL would...”, the word “**the**” has been deleted.

5. Page 3

Original text: To address this issue, the strategy of doping two different emission species (like as $\text{Eu}^{2+}/\text{Eu}^{3+}$ 16, $\text{Pr}^{3+}/\text{Tb}^{3+}$ 17, $\text{Mn}^{4+}/\text{Eu}^{3+}$ 18,19, $\text{Ce}^{3+}/\text{Mn}^{4+}$ 20, etc.) into suitable compounds has been developed, because it has the potential to simultaneously achieve high sensitivity and excellent signal discriminability in temperature sensing.

Suggested modification: To address this issue, the strategy of doping two different emission species (like as $\text{Eu}^{2+}/\text{Eu}^{3+}$ 16, $\text{Pr}^{3+}/\text{Tb}^{3+}$ 17, $\text{Mn}^{4+}/\text{Eu}^{3+}$ 18,19, $\text{Ce}^{3+}/\text{Mn}^{4+}$ 20, etc.) into suitable compounds has been developed, because it has the potential to simultaneously achieve high sensitivity and excellent signal discrimination in temperature sensing.

Author Reply: On page 3, the sentence “.., because it has the potential to simultaneously achieve high sensitivity and excellent signal discriminability in temperature sensing.” has been corrected as “... **because** it has the potential **to simultaneously** achieve high sensitivity and excellent signal discrimination in temperature sensing.”

5. Page 4

Original text: Experimental evidence from extended X-ray absorption fine structure, and electron diffraction, as well as density functional theory (DFT) calculations, unambiguously suggests that

the green and red emissions in $\text{Li}_2\text{ZnSiO}_4:\text{Mn}^{2+}$ come from the isolated Mn^{2+} ions and $\text{Mn}^{2+}-\text{Mn}^{2+}$ dimers, respectively.

Suggested modification: Experimental evidence from extended X-ray absorption fine structure and electron diffraction, as well as density functional theory (DFT) calculations, unambiguously suggest that the green and red emissions in $\text{Li}_2\text{ZnSiO}_4:\text{Mn}^{2+}$ come from the isolated Mn^{2+} ions and $\text{Mn}^{2+}-\text{Mn}^{2+}$ dimers, respectively.

Author Reply: On page 4, in the sentence “Experimental ..., unambiguously suggests that ...”, the word “suggests” has been changed to “**suggest**”.

6. Page 5

Original text: The E_f of substitution model M3 (-0.89 eV) is significantly smaller than that of M1 (6.56 eV) or M2 (6.80 eV) (see Fig. 1b), which firmly confirmed that at a low Mn^{2+} concentration, Mn^{2+} occupied the Zn^{2+} site in this structure.

Suggested modification: The E_f value of substitution model M3 (-0.89 eV) is significantly smaller than that of M1 (6.56 eV) or M2 (6.80 eV) (see Fig. 1b), which firmly confirmed that at a low Mn^{2+} concentration, Mn^{2+} occupied the Zn^{2+} site in this structure.

Author Reply: on page 5, the sentence “The E_f of ...” has been changed to “The E_f **value** of substitution ...”.

7. Page 5

Original text: Based on this result, we further calculated the E_f as the second Mn (Mn2) was introduced in substitution model M3 ($\text{Li}_8\text{Zn}_3\text{MnSi}_4\text{O}_{16}$).

Suggested modification: Based on this result, we further calculated E_f as the second Mn (Mn2) was introduced in the substitution model M3 ($\text{Li}_8\text{Zn}_3\text{MnSi}_4\text{O}_{16}$).

Author Reply: On page 5, the sentence “...was introduced in substitution model M3 ($\text{Li}_8\text{Zn}_3\text{MnSi}_4\text{O}_{16}$).” has been modified as “... was introduced in **the** substitution model M3 ($\text{Li}_8\text{Zn}_3\text{MnSi}_4\text{O}_{16}$).”.

8. Page 5

Original text: The E_f of M7(-3.65 eV) is much lower than that of others, suggesting that the second Mn^{2+} substituted Li2, and the substituted Li2 replaced Zn^{2+} to maintain the charge balance.

Suggested modification: The E_f value of M7(-3.65 eV) is much lower than that of others, suggesting that the second Mn^{2+} substituted Li2, and the substituted Li2 replaced Zn^{2+} to maintain the charge balance.

Author Reply: On page 5, the sentence “The E_f of M7(-3.65 eV) is....” has been changed to “The E_f **value** of M7(-3.65 eV) is ...”

9. Page 6

Original text: Under this substitution model (M7), Mn1 and Mn2 have a distance of 1.87 Å and share with one oxygen, which suggests that $\text{Mn}^{2+}(\text{Zn})\text{-Mn}^{2+}(\text{Li}_2)$ dimer can be formed in relatively high-concentration in Mn^{2+} -ions-doped samples.

Suggested modification: Under this substitution model (M7), Mn1 and Mn2 have a distance of 1.87 Å and share with one oxygen, which suggests that $\text{Mn}^{2+}(\text{Zn})\text{-Mn}^{2+}(\text{Li}_2)$ dimer can be formed at relatively high-concentrations in Mn^{2+} -ions-doped samples.

Author Reply: On page 6, the sentence "...can be formed in relatively high-concentration in Mn^{2+} -ions-doped samples" has been changed to "... can be formed **at** relatively high-concentrations in Mn^{2+} -ions-doped samples."

10. Page 6

Original text: The XANES spectra of $\text{Li}_2\text{Zn}_{0.95}\text{SiO}_4:0.05\text{Mn}^{2+}$ and $\text{Li}_2\text{Zn}_{0.85}\text{SiO}_4:0.15\text{Mn}^{2+}$ are considerably similar to that of MnO than those of Mn_2O_3 or MnO.

Suggested modification: The XANES spectra of $\text{Li}_2\text{Zn}_{0.95}\text{SiO}_4:0.05\text{Mn}^{2+}$ and $\text{Li}_2\text{Zn}_{0.85}\text{SiO}_4:0.15\text{Mn}^{2+}$ are considerably similar to that of MnO than those of Mn_2O_3 or MnO_2 .

Author Reply: On page 6, the sentence "...are considerably similar to ..." has been corrected as "...are **considerably** similar to that of MnO than those of Mn_2O_3 or **MnO₂**."

11. Page 6

Original text: The detailed fitting parameters are summarised Supplementary Table 1.

Suggested modification: The detailed fitting parameters are summarized in Supplementary Table 1.

Author Reply: On page 6, the sentence "...are summarised Supplementary Table 1." has been changed to "...are **summarized in** Supplementary Table 1.". Because the EXAFS analysis part has been moved to the supporting part, the corresponding modifications has been added in the supporting information part (Page 6).

12. Page 7

Original text: Fig. 2a shows a high-resolution transmission electron microscope (HR-TEM) image of a $\text{Li}_2\text{Zn}_{0.95}\text{SiO}_4:0.05\text{Mn}^{2+}$ particle, in which the interplanar distances are determined as 0.52 and 0.38 nm, corresponding to the (020) and (101) lattice planes of $\text{Li}_2\text{ZnSiO}_4$, respectively.

Suggested modification: Fig. 2a shows a high-resolution transmission electron microscopy (HR-TEM) image of an $\text{Li}_2\text{Zn}_{0.95}\text{SiO}_4:0.05\text{Mn}^{2+}$ particle, in which the interplanar distances are determined as 0.52 and 0.38 nm, corresponding to the (020) and (101) lattice planes of $\text{Li}_2\text{ZnSiO}_4$, respectively.

Author Reply: Thanks. Because the TEM characterization of $\text{Li}_2\text{Zn}_{0.95}\text{SiO}_4:0.05\text{Mn}^{2+}$ has been moved to the supporting information part (Supplementary Figure 4), the related descriptions have been deleted.

13. Page 7

Original text: Figs. 2g and h further show the HR-TEM images of a $\text{Li}_2\text{Zn}_{0.85}\text{SiO}_4:0.15\text{Mn}^{2+}$ particle.

Suggested modification: Figs. 2g and 2h further show the HR-TEM images of a $\text{Li}_2\text{Zn}_{0.85}\text{SiO}_4:0.15\text{Mn}^{2+}$ particle.

Author Reply: On page 7, the “Figs. 2g and h” has been changed to “Figs. 2g and 2h”. Because a new Figure 2 has been provided in the revised manuscript, the “Figs. 2g and 2h further show the HR-TEM images of a $\text{Li}_2\text{Zn}_{0.85}\text{SiO}_4:0.15\text{Mn}^{2+}$ particle” has been changed to “**Fig. 2a shows the HR-TEM image of a $\text{Li}_2\text{Zn}_{0.85}\text{SiO}_4:0.15\text{Mn}^{2+}$ particle.**”.

14. Page 7

Original text: Specifically, some additional weak points (highlighted with red and purple line circles) can be observed in Figs. 2i and j.

Suggested modification: Specifically, some additional weak points (highlighted with red and purple line circles) can be observed in Figs. 2i and 2j.

Author Reply: On page 7, the “Figs. 2i and j.” has been changed to “**Figs. 2i and 2j.**”. Since a new Figure 2 has been provided in the revised manuscript, the , the “Figs. 2i and 2j.” has been corrected as “**Figs. 2b and 2c.**”. Thanks.

15. Page 9

Original text: When the doping concentration is further increased to $x = 0.30$, no significant change is found in the excitation spectrum compared with the above samples, implying that both emission bands originated from the Mn^{2+} species.

Suggested modification: When the doping concentration is further increased to $x = 0.30$, no significant change is found in the excitation spectrum compared with the above samples, implying that both emission bands originated from the Mn^{2+} species.

Author Reply: On page 9, in this sentence, the words “significant” and “origianted” have been corrected as “**significant**” and “**originated**”, respectively.

16. Page 9

Original text: The luminescence decay curves of the two emission bands are provided in Figs. 3c and d, respectively.

Suggested modification: The luminescence decay curves of the two emission bands are provided in Figs. 3c and 3d, respectively.

Author Reply: On page 9, the “Figs. 3c and d” has been changed to “Figs. 3c and **3d**”. Because the Figs. 3c and 3d have been moved to the supporting information part (Supplementary Figure 7), the related description has been deleted. Thanks.

17. Page 10

Original text: The monotonous decrease in the emission intensities of the two emission bands with increasing temperature is mainly ascribed to the gradually increasing nonradiative processes.

Suggested modification: The monotonous decrease in the emission intensities of the two emission bands with increasing temperature is mainly ascribed to the gradually increasing nonradiative processes.

Author Reply: On page 10, the words “intensities” and “mainly” have been modified as “**intensities**” and “**mainly**”, respectively. However, to simplify this manuscript, this sentence has been deleted in the revised manuscript. Thanks.

18. Page 10

Original text: A relatively high temperature can directly increase the Mn^{2+} - Mn^{2+} distance, which will reduce the number of effective Mn^{2+} - Mn^{2+} dimer³².

Suggested modification: A relatively high temperature can directly increase the Mn^{2+} - Mn^{2+} distance, which will reduce the number of effective Mn^{2+} - Mn^{2+} dimers³².

Author Reply: On page 10, the sentence “....., which will reduce the number of effective Mn^{2+} - Mn^{2+} dimer³².” has been changed to “....., which will reduce the number of effective Mn^{2+} - Mn^{2+} **dimers**³².”.

19. Page 10

Original text: Consequently, the red emission from the dimers has worse thermal stability than that the green one (from the isolated Mn^{2+} ions).

Suggested modification: Consequently, the red emission from the dimers has worse thermal stability than that of the green one (from the isolated Mn^{2+} ions).

Author Reply: On page 10, the sentence “Consequently, the red emission from the dimers has worse thermal stability than that the green one (from the isolated Mn^{2+} ions).” has been modified as “Consequently, the red emission from the dimers has worse thermal stability than that **of the** green one (from the isolated Mn^{2+} ions).”.

20. Page 11

Original text: Because of the similar excitation spectra and different thermal quenching behaviours of the two emission bands, the promising application of $\text{Li}_2\text{Zn}_{0.85}\text{SiO}_4:0.15\text{Mn}^{2+}$ in wearable temperature sensors has been investigated and demonstrated.

Suggested modification: Because of the similar excitation spectra and different thermal quenching behaviors of the two emission bands, the promising application of $\text{Li}_2\text{Zn}_{0.85}\text{SiO}_4:0.15\text{Mn}^{2+}$ in wearable temperature sensors was investigated.

Author Reply: On page 11, in the sentence of “Because of the similar excitation spectra and different thermal quenching behaviours of ..., the promising application...”, the word “behaviours” has been changed to “**behaviors**” in the revised manuscript. Additionally, the description “has been investigated and demonstrated” has been changed to “**was investigated**”.

21. Page 11

Original text: To acquire good light guiding and sensing performance of the sensor, a transparent optical encapsulant (OE) and polydimethylsiloxane (PDMS) with a high refractive index (RI) difference have been selected as the matrix materials to fabricate flexible optical fibre38, and the designed fibre with a double-cladding and double-tail fibre structure is provided in Fig. 4a.

Suggested modification: To acquire good light guiding and sensing performance of the sensor, a transparent optical encapsulant (OE) and polydimethylsiloxane (PDMS) with a high refractive index (RI) difference were selected as the matrix materials to fabricate flexible optical fibre38, and the designed fibre with a double-cladding and double-tail fibre structure is provided in Fig. 4a.

Author Reply: On page 11, in the sentence “To acquire good in Fig. 4a”, the words “have been selected as” has been changed to “**were** selected as” in the revised manuscript.

22. Page 11

Original text: In this fibre, the core was fabricated using an OE with a high refractive index ($n_1 = 1.53$), which has excellent light guiding properties. The inner cladding was fabricated by mixing the OE with $\text{Li}_2\text{Zn}_{0.85}\text{SiO}_4:0.15\text{Mn}^{2+}$ powder consisting of irregular particles with a size of $\sim 1\text{--}5.5\ \mu\text{m}$ (Supplementary Fig. 5), and the inner cladding acted as the temperature-sensing fluorescence response layer.

Suggested modification: In this fibre, the core was fabricated using OE with a high refractive index ($n_1 = 1.53$), which has excellent light guiding properties. The inner cladding was fabricated by mixing OE with $\text{Li}_2\text{Zn}_{0.85}\text{SiO}_4:0.15\text{Mn}^{2+}$ powder consisting of irregular particles with a size of $\sim 1\text{--}5.5\ \mu\text{m}$ (Supplementary Fig. 5), and the inner cladding acted as the temperature-sensing fluorescence response layer.

Author Reply: On page 11, the sentence “...was fabricated using an OE with a high refractive index ($n_1 = 1.53$)....” has been modified as “...was fabricated using OE with a high refractive

index ($n_1 = 1.53$)...”. Moreover, in the sentence of “The inner cladding, and the inner cladding acted”, the word “cladding” has been corrected as “**cladding**”.

23. Page 11

Original text: The silica fibres were used to guid the of excitation light and collect fluorescence.

Suggested modification: The silica fibres were used to guide the excitation light and collect fluorescence.

Author Reply: On page 11, the sentence “... were used to guid the of excitation light and collect fluorescence.” has been corrected as “... were used to **guide the excitation light** and collect fluorescence.”.

24. Page 12

Original text: The numerical aperture ($NA = 2221n - n$) of the designed optical fibre is about 0.59, exhibiting a good ability to collect and guid light.

Suggested modification: The numerical aperture ($NA = 2221n - n$) of the designed optical fibre is about 0.59, exhibiting a good ability to collect and guide light.

Author Reply: On page 12, in the sentence “The number apertureand guid light”, the word “guid” has been corrected as “**guide**” in the revised manuscript. Since the fabrication procedure of the optical fibre has been moved to the supporting information part, the related modification has been made in the revised supporting part.

25. Page 12

Original text: When an LED with a central wavelength of 365 nm UV-light was coupled into this flexible sensor through pigtailed silica optical fibre with a core diameter of 200 μm , yellow-green fluorescence from the optical fibre can be observed, corresponding to the emission of $\text{Li}_2\text{Zn}_{0.85}\text{O}_4:0.15\text{Mn}^{2+}$ (Fig. 4h).

Suggested modification: When an LED device with a central wavelength of 365 nm UV-light was coupled into this flexible sensor through pigtailed silica optical fibre with a core diameter of 200 μm , yellow-green fluorescence from the optical fibre can be observed, corresponding to the emission of $\text{Li}_2\text{Zn}_{0.85}\text{O}_4:0.15\text{Mn}^{2+}$ (Fig. 4h).

Author Reply: On page 12, in the sentence “When an LED with a central wavelength....., (Fig. 4h)” has been changed to “When an LED **device** with a central wavelength....., (Fig. 4h)”. Besides, owing to a new Figure 4 is provided in the revised manuscript, the “Fig. 4h” has been changed to “**Fig. 4g**”.

26. Page 13

Original text: As shown in Fig. 5c, the sensor is calibrated in the range of -20 to 100 °C (each data point was measured three times to take the average value), showing good parabola linearity, and it well-fitted: $I_{530} / I_{650} = 2.3483 \times 10^{-5} T^2 + 4.9851 \times 10^{-4} T + 0.3282$.

Suggested modification: As shown in Fig. 5c, the sensor is calibrated in the range of -20 to 100 °C (each data point was measured three times to take the average value), showing good parabola linearity, and it is well-fitted by relation: $I_{530} / I_{650} = 2.3483 \times 10^{-5} T^2 + 4.9851 \times 10^{-4} T + 0.3282$.

Author Reply: On page 13, in the sentence “As shown in Fig. 5c, the sensor is, showing good, and it well fitted:.....” has been changed to “As shown in Fig. 5c, the sensor is, showing good..., and it is well fitted:.....”.

27. Page 13

Original text: The temperature response of the optical fibre sensor in the metal heater and the in-situ placed thermistor have been tested as shown in Fig. 5g.

Suggested modification: The temperature response of the optical fibre sensor in the metal heater and the in-situ placed thermistor have been tested, as shown in Fig. 5g.

Author Reply: On page 13, the sentence “... have been tested as shown in Fig. 5g.” has been changed to “... have been tested, **as shown in** Fig. 5g.”. Since a new Figure 5 has been provided in the revised manuscript, the related description has been changed to “, **as shown in Fig. 5h**.”.

28. Page 14

Original text: In summary, a series of Mn^{2+} doped Li_2ZnSiO_4 phosphors have been synthesized by a facile solid-state reaction method.

Suggested modification: In summary, a series of Mn^{2+} doped Li_2ZnSiO_4 phosphors was synthesized by a facile solid-state reaction method.

Author Reply: on page 14, in the sentence “a series of Mn^{2+} doped Li_2ZnSiO_4 phosphors have been synthesized by a facile solid-state reaction method.”, the words “have been” have been changed to “**was**”.

29. Page 14

Original text: Benefiting from the similar excitation spectra and different thermal quenching of the two emission bands, a stable flexible and wearable optical fibre temperature sensor was fabricated based on the luminescence material $Li_2Zn_{0.85}SiO_4:0.15Mn^{2+}$ and some polymers.

Suggested modification: Benefiting from the similar excitation spectra and different thermal quenching behaviors of the two emission bands, a stable flexible and wearable optical fibre

temperature sensor was fabricated based on the luminescence material $\text{Li}_2\text{Zn}_{0.85}\text{SiO}_4:0.15\text{Mn}^{2+}$ and some polymers.

Author Reply: On page 14, in the sentence "...thermal quenching behaviours of the two emission bands, a stable flexible and wearable optical fibre temperature sensor was fabricated based on the luminescence material $\text{Li}_2\text{Zn}_{0.85}\text{SiO}_4:0.15\text{Mn}^{2+}$ and some ploymers.", the words "behaviours", "ploymers" and "optical" have been corrected as "**behaviors**", "**polymers**" and "**optical**", respectively.

30. Page 14

Original text: Additonally, the optical fibre temperature sensor shows good performances in real-time contact and non-contract temperature measurements, and which has good potential for monitoring of human thermal activities.

Suggested modification: Additionally, the optical fibre temperature sensor shows good performances in real-time contact and non-contact temperature measurements, and it has good potential for monitoring of human thermal activities.

Author Reply: On page 14, in the sentence "Additonally, the optical fibre temperature sensor shows good performances in real-time contact and non-contract temperature measurements, and which has good potential for monitoring of human thermal activities.", the word "Additonally" has been corrected as "**Additionally**" and the word "which" has been changed to "**it**".

31. Page 15

Original text: Phosphor samples $\text{Li}_2\text{Zn}_{1-x}\text{SiO}_4:x\text{Mn}^{2+}$ ($x = 0.01-0.30$) are synthesized viay a conventional solid state reaction method and the raw materials are Li_2CO_3 (99.99%), $\text{SiO}_2(\text{AR})$, $\text{ZnO}(99.9\%)$ and $\text{MnCO}_3(99.9\%)$, and all the raw materials are used as received without further purification.

Suggested modification: Phosphor samples $\text{Li}_2\text{Zn}_{1-x}\text{SiO}_4:x\text{Mn}^{2+}$ ($x = 0.01-0.30$) are synthesized via a conventional solid state reaction method and the raw materials are Li_2CO_3 (99.99%), $\text{SiO}_2(\text{AR})$, $\text{ZnO}(99.9\%)$ and $\text{MnCO}_3(99.9\%)$, and all the raw materials are used as received without further purification.

Author Reply: On page 15, in the sentence "Phosphor samples..... viay, further purification.", the word "viay" has been corrected as "**via**".

32. Besides purity, supplier should be reported for each raw material.

Author Reply: Thanks. Both the purity and supplier have been added in the revised manuscript. Li_2CO_3 (99.99% **metals basis**), $\text{SiO}_2(\text{AR}, 99\%)$, $\text{ZnO}(99.9\% \text{ **metals basis**)}$ and $\text{MnCO}_3(99.9\%$

metals basis). All the raw materials were purchased from Aladdin Industrial Corporation (Shanghai, China) and used as received without further purification.

33. Page 15

Original text: The raw materials are weighted according to the nominal composition $\text{Li}_2\text{Zn}_{1-x}\text{SiO}_4:\text{xMn}^{2+}$, then ground and mix it well in a mortar.

Suggested modification: The raw materials are weighted according to the nominal composition $\text{Li}_2\text{Zn}_{1-x}\text{SiO}_4:\text{xMn}^{2+}$, then ground and mixed well in a mortar.

Author Reply: On page 15, in the sentence “The raw materials are weighted according to the nominal composition $\text{Li}_2\text{Zn}_{1-x}\text{SiO}_4:\text{xMn}^{2+}$, then ground and mix it well in a mortar.”, the word “it” has been deleted and the “mix” has been changed to “mixed” in the revised manuscript.

34. Page 15

Original text: The morphology characterizations were measured by using the SEM (Nova, NANO SEM 430) and TEM (JEOL, 2100F).

Suggested modification: The morphology characterizations were measured by using SEM (Nova, NANO SEM 430) and TEM (JEOL, 2100F) methods.

Author Reply: On page 15, the sentence “The morphology characterizations were measured by using the SEM (Nova, NANO SEM 430) and TEM (JEOL, 2100F).” has been changed to “The morphology characterizations were measured by using SEM (Nova, NANO SEM 430) and TEM (JEOL, 2100F) methods.”.

35. Page 15

Original text: The luminescence thermal quenching behavior and photoluminescence quantum yield (QY) of the sample are measured by the same spectrofluorimeter which are equipped with a TAP-02 High- temperature fluorescence instrument (Tian Jin Orient – KOJI instrument Co., Ltd.) and an integrated sphere, respectively.

Suggested modification: The luminescence thermal quenching behavior and photoluminescence quantum yield (QY) of the sample are measured by the same spectrofluorimeter which is equipped with a TAP-02 High- temperature fluorescence instrument (Tian Jin Orient – KOJI instrument Co., Ltd.) and an integrated sphere.

Author Reply: On page 15, the sentence “...the sample are measured by the same spectrofluorimeter which are equipped ...” has been changed to “... the sample is measured by the same spectrofluorimeter which is equipped with a TAP-02 High- temperature fluorescence instrument (Tian Jin Orient – KOJI instrument Co., Ltd.).”.

Reviewer #2 (Remarks to the Author): The Authors report possibility to use fluorescence intensity ratio (FIR) material to develop wearable optical temperature sensor. FIR materials are hot topic with big amount of publications and interesting possibilities. The Authors give theoretical background and details about FIR material LiZnSiO:Mn, but the main result is usage of LiZnSiO:Mn for wearable health monitoring. The Authors prepare flexible optical fiber with LiZnSiO:Mn in inner core of the fiber. Specially during time of Covid-19, Fig. 5f is extremely interesting.

Author Reply: Thanks for your valuable comments and suggestion.

1. The problem in the present manuscript is that the main part, developing of optical fiber for wearable temperature sensing, is only small part of the text. Main part is analysis of LiZnSiO:Mn. To what extend SEM, TEM, XANES, EXAFS, DFT and Vasp are crucial for the main result? Or should part of them be separated to separate manuscript or to Supplementary Information? These results are not mentioned in Abstract. The present form of the manuscript is rather long.

Author Reply: To make this manuscript more compact, TEM of the $\text{Li}_2\text{Zn}_{0.95}\text{SiO}_4:0.05\text{Mn}^{2+}$, EXAFS, luminescence decay curves of the samples, and the fibre fabrication procedure have been moved to the Supplementary Information part. At the same time, the temperature sensitivity (S_d/S_r), precision, and cytotoxicity analysis of the optical fiber have been added in the revised manuscript. Based on above changes, the abstract has been modified as following:

Photothermal sensing is crucial for the creation of smart wearable device. However, the discovery of luminescent materials with suitable dual-wavelength emissions is a great challenge for the construction of stable wearable optical fibre temperature sensors. Benefiting from the Mn^{2+} - Mn^{2+} super-exchange interactions, a dual-wavelength (530/650 nm) emitting material $\text{Li}_2\text{ZnSiO}_4:\text{Mn}^{2+}$ is presented via simple increasing the Mn^{2+} concentration, wherein the two emission bands have different temperature-dependent emission behaviors, but exhibit quite similar excitation spectra. Density functional theory calculations, coupled with extend X-ray absorption fine structure and electron-diffraction analyses reveal the origins of the two emission bands in this material. A wearable optical temperature sensor is fabricated by incorporating $\text{Li}_2\text{ZnSiO}_4:\text{Mn}^{2+}$ in stretchable elastomers based-optical fibres, which can provide thermal-sensitive emissions at dual-wavelengths for stable ratiometric temperature sensing with good precision and repeatability. More importantly, a wearable mask integrated with this stretchable fibre sensor is demonstrated for the detection of physiological thermal changes, showing great potential for use as a wearable health monitor. This study also provides a new framework for creating novel transition-metal-activated luminescence materials.

2. If the Authors analyze LiZnSiO:Mn, then they should also analyze which material would be most suitable for optical fiber. Why LiZnSiO:Mn? At present time exist rather big selection of FIR materials.

Author Reply: As mentioned in the introduction part, various dual-wavelength emitting materials have been developed for ratio-metric temperature sensing. The dual-wavelength emitting materials based on two different emission species, such as $\text{Eu}^{2+}/\text{Eu}^{3+}$ (Y. Pan, et al., *Adv. Mater.* 2018, 30, 1705256.), $\text{Pr}^{3+}/\text{Tb}^{3+}$ (Y. Wu, et al., *Inorg. chem. Front.* 2008, 5, 2456.), $\text{Mn}^{4+}/\text{Eu}^{3+}$ (P. Wang, et al., *Dalton Trans.* 2019, 48, 10062.; Z. Long, et al., *ACS Appl. Electron. Mater.* 2020, 2, 3889.), $\text{Ce}^{3+}/\text{Mn}^{4+}$ (Y. Chen, et al., *Inorg. Chem.* 2020, 59, 1383.) etc., co-doping were gaining increasing attention for their potential to simultaneously achieve high sensitivity and excellent signal discriminability in temperature sensing. However, since the two emission bands of these materials possess different excitation spectra, their FIR values are affected by the wavelength of the excitation light source as well, which may affect the stability of the related temperature sensor. The $\text{Li}_2\text{ZnSiO}_4:\text{Mn}^{2+}$ exhibits dual-wavelength emission with different thermal quenching behaviors, but the two emission bands display quite similar excitation spectra. Furthermore, a stable optical fibre temperature sensor can be expected by fabricating this material in fibre. Therefore, the material $\text{Li}_2\text{ZnSiO}_4:\text{Mn}^{2+}$ is more suitable for temperature sensor as compared to the previous dual-wavelength emission materials.

3. Usually in FIR analyses is give value for absolute and relative temperature sensitivity, S_a and S_r . Now the Authors give only reproducibility (Fig. 5d).

Author Reply: Thanks for your suggestion. This sensor can be calibrated in the range from -20°C to 100°C , showing good parabola linearity, and it was well-fitted as following:

$$FIR = I_{530} / I_{650} = 2.348 \times 10^{-5} T^2 + 4.985 \times 10^{-4} T + 0.328 \quad (1)$$

The absolute sensitivity (S_a) and relative sensitivity (S_r) of the fibre were calculated by the following equations:

$$S_a = \frac{dFIR}{dT} = \frac{d(I_{530} / I_{650})}{dT} \quad (2)$$

$$S_r = \left| \frac{1}{FIR} \frac{dFIR}{dT} \right| = \left| \frac{1}{(I_{530} / I_{650})} \frac{d(I_{530} / I_{650})}{dT} \right| \quad (3)$$

According to above equations and Figure 5c, the maximal sensitivity of S_a and S_r values of the

sensor were determined as $0.0052\text{ }^{\circ}\text{C}^{-1}$ and $0.848\%\text{ }^{\circ}\text{C}^{-1}$ at $100\text{ }^{\circ}\text{C}$, as shown in Supplementary Figure 11a. Considering the sensor will be used in the wearable field, we further measured the S_a and S_r of the sensor around the body temperature range. The FIR values of the sensor at the temperature range of 34 to $44\text{ }^{\circ}\text{C}$ were provided in supplementary Fig. 11b. It is observed that the temperature dependent FIR values can be fitted and simplified approximately with a linear equation: $y = 0.00257T + 0.2817$ (see Supplementary Figure 11c). Therefore, the S_a and S_r values at the temperature range of 34 - $44\text{ }^{\circ}\text{C}$ were calculated as $0.00257\text{ }^{\circ}\text{C}^{-1}$ and $0.682\%\text{ }^{\circ}\text{C}^{-1}$, respectively, which are moderate and acceptable values in temperature sensor application. Moreover, these values of the sensor are comparable to previous reported electrical sensors (T. Q. Trung, et al., *ACS Appl. Mater. Interfaces* 2018, 11, 2317.) and the rare-earth ions doped silica optic fibre sensors (D. Manzani, et al., *Sci. Rep.* 2017, 7, 41596.).

Action: The S_a and S_r of the temperature sensor have been added in the revised manuscript. Besides, some necessary modifications have been made in the manuscript and the supporting information part (Supplementary Figure 11).

Supplementary Figure 11 a The S_a and S_r values of the temperature sensor as a function of the measure temperature according to Fig. 5b. **b** The fitting curve of the emission ratios of green (530 nm) to red emission (650 nm) based on Fig. 5b and the measured emission ratios of green (530 nm) to red emission (650 nm) with an interval of $2\text{ }^{\circ}\text{C}$ (red circle). **c** The linear fitting curve of the

emission ratio of the green (530 nm) to red emission (650 nm) and the corresponding experimental data according to Supplementary Figure 11b.

4. Small problem in the manuscript is that it is not very carefully written. On line 49 is written because, line 64 simultaneously, on line 136 should read MnO₂ and on line 172 should read 2i and k, on lines 183 and 188 is green emission at 525 and 530 nm (which is correct?), on line 205 a line should be replaced by a comma, on line 231 word lower should be removed, on line 244 is word intensities, on line 245 is word mainly, on line 265 is word caldding, line 320 and Fig. 5c contains too many decimals, on line 348 is word optical, on line 361 is word viay, on line 407 mjuu is missing, and on line 444 concentration 005 should be replaced by 015.

Author Reply: Thanks for your very careful review and valuable suggestion. All the issues have been modified carefully and highlighted in the revised manuscript.

5. Most of the figures have too small fonts.

Author Reply: Thanks. We have made the modification accordingly in the revised manuscript.

6. In Fig. 2b scale bar is not clearly visible.

Author Reply: Fig. 2b is the Fourier-transform diffraction patterns of the Li₂Zn_{0.95}SiO₄:0.05Mn²⁺ particle viewed along [101] zone axis taken from the HR-TEM in Fig. 2a, which usually does not require a scale bar. However, the scale bar in other places of Figure 2 and Supplementary Figure 4 has been enlarged.

Fig. 2. TEM characterization of $\text{Li}_2\text{ZnSiO}_4:\text{Mn}^{2+}$. **a** HR-TEM image of a $\text{Li}_2\text{Zn}_{0.85}\text{SiO}_4:0.15\text{Mn}^{2+}$ particle. The inset shows the low-resolution TEM image of a $\text{Li}_2\text{Zn}_{0.85}\text{SiO}_4:0.15\text{Mn}^{2+}$ particle. **b** Fourier-transform diffraction patterns of the $\text{Li}_2\text{Zn}_{0.85}\text{SiO}_4:0.15\text{Mn}^{2+}$ particle viewed along $[\bar{3}10]$ zone axis taken from the HR-TEM in a) (red-dot highlighted rectangular area). **c** Selected area electron diffraction pattern of the $\text{Li}_2\text{Zn}_{0.85}\text{SiO}_4:0.15\text{Mn}^{2+}$. **d,f** the simulated electron diffraction pattern and the corresponding crystallographic model of M7 viewed along $[\bar{3}10]$ zone axis. **e,g** The simulated electron diffraction pattern and the corresponding crystallographic model of $\text{Li}_2\text{ZnSiO}_4$ viewed along $[\bar{3}10]$ zone axis.

Supplementary Figure 4 a HR-TEM image of a $\text{Li}_2\text{Zn}_{0.95}\text{SiO}_4:0.05\text{Mn}^{2+}$ particle. The inset shows the low-resolution TEM image of the $\text{Li}_2\text{Zn}_{0.95}\text{SiO}_4:0.05\text{Mn}^{2+}$ particle. **b** Fourier-transform diffraction patterns of the $\text{Li}_2\text{Zn}_{0.95}\text{SiO}_4:0.05\text{Mn}^{2+}$ particle viewed along $[\bar{1}01]$ zone axis taken from the HR-TEM in a). **c,e** Simulated electron diffraction pattern and the corresponding crystallographic model of $\text{Li}_2\text{ZnSiO}_4$ viewed along $[\bar{1}01]$ zone axis. **d,f** The simulated electron diffraction pattern and the corresponding crystallographic model of M7 viewed along $[\bar{1}01]$ zone axis.

7. Figs 2g and 2h looks similar,

Author Reply: Fig. 2h is the enlarged view of Fig. 2g so that they look similar. To avoid confusing, Fig. 2g has been deleted in the revised manuscript (see above Fig. 2).

8. Em 530 nm (green line) in Fig. 3b is not visible.

Author Reply: I am sorry for the issue. To clearly show the Em 530 nm (green line), some modifications have been made in Fig. 3b.

9. Finally, the title could be more compact. At least, three first words (Doping Concentration-Controlled) are not needed, because main result is not the concentration analysis.

Author Reply: Thanks for your suggestion. The title “Doping Concentration-Controlled Dual-Wavelength Emitting Materials toward Wearable Optical Fibre Temperature Sensor” has been changed to “**Mn²⁺-activated dual-wavelength emitting materials toward wearable optical fibre temperature sensor**”.

10. The manuscript is interesting and it will attract big amount of attention, but it should be written more carefully and in more compact form.

Author Reply: Thanks for your valuable suggestion. We have carefully revised the whole manuscript to make this manuscript more compact. All the changes are highlighted in the revised manuscript.

Reviewer #3 (Remarks to the Author): The author reported a novel dual-wavelength emitting material $\text{Li}_2\text{ZnSiO}_4:\text{Mn}^{2+}$ as wearable optical fiber temperature sensors. It is interesting that the two emission bands (525/650 nm) of the material have different concentration quenching and temperature quenching behaviors, but exhibit similar excitation spectra. Based on DFT calculations, EXAFS and electron diffraction measurements, the author clearly demonstrated that the green and red emissions were ascribed to the isolated Mn^{2+} and $\text{Mn}^{2+}\text{-Mn}^{2+}$ dimer, respectively. It is crucial for the understanding and design of the luminescence behavior and the luminescence tuning of transition metal Mn^{2+} activated phosphors. Moreover, based on the new fiber fabrication procedure, the author extended the application of Mn^{2+} doped phosphors to the wearable optical fiber temperature sensor, showing high promise in human health monitoring. Overall, this is a well-prepared manuscript with an innovative demonstration for me. This paper is unique and may be publishable with minor revisions. My concerns are summarized in the following.

1. The corresponding lattice planes of the extra electron diffraction points in Figure 2k,i should be identified and marked.

Author Reply: Thanks for your suggestion. Since the lattice planes of the extra electron diffraction points in previous Fig. 2k and 2i are the same, we just added the corresponding lattice planes of the extra electron diffraction points in Figure 2k (see Figure 2b). Besides, a new Figure 2 and some necessary modifications have been provided in the revised manuscript.

Fig. 2 a HR-TEM image of a $\text{Li}_2\text{Zn}_{0.85}\text{SiO}_4:0.15\text{Mn}^{2+}$ particle. The inset shows the low-resolution TEM image of a $\text{Li}_2\text{Zn}_{0.85}\text{SiO}_4:0.15\text{Mn}^{2+}$ particle. b Fourier-transform diffraction patterns of the $\text{Li}_2\text{Zn}_{0.85}\text{SiO}_4:0.15\text{Mn}^{2+}$ particle viewed along $[3\bar{1}0]$ zone axis taken from the HR-TEM in a) (red-dot highlighted rectangular area). c Selected area electron diffraction pattern of the $\text{Li}_2\text{Zn}_{0.85}\text{SiO}_4:0.15\text{Mn}^{2+}$. d,f the simulated electron diffraction pattern and the corresponding crystallographic model of M7 viewed along $[3\bar{1}0]$ zone axis. e,g The simulated electron diffraction

pattern and the corresponding crystallographic model of $\text{Li}_2\text{ZnSiO}_4$ viewed along $[\bar{3}10]$ zone axis.

2. The introductions of the insets in Figure 5e,f should be added in the corresponding Figure caption part.

Author Reply: Thanks for your suggestion. The related information has been added in the revised manuscript.

3. As shown in Figure 1a, the crystal structure of $\text{Li}_2\text{ZnSiO}_4$ is provided, while no reference or card number (ICSD? or others) about this structure is shown in the manuscript. The information about the source of the crystal structure should be given.

Author Reply: Thanks for your suggestion. The ICSD card number of $\text{Li}_2\text{ZnSiO}_4$ (ICSD no. 8237) has been added in the revised manuscript.

4. Figure 3b, only relatively weak excitation at ~ 365 nm of the phosphor $\text{Li}_2\text{ZnSiO}_4:\text{Mn}^{2+}$ is shown, but why the authors used the 365 nm UV-LED as the excitation source in the fabricated wearable optical fiber temperature sensor?

Author Reply: Indeed, the 365 nm is not the strongest excitation position for the phosphor sample. However, upon 365 nm UV-light excitation, one can clearly observe the emission from $\text{Li}_2\text{ZnSiO}_4:\text{Mn}^{2+}$ powder (Supplementary Figure 6) and fibre fabricated in current work (Figure 4h), indicating that using the 365 nm UV-LED as excitation is reasonable.

5. As shown in Figure 3b, the two emission bands of $\text{Li}_2\text{ZnSiO}_4:\text{Mn}^{2+}$ have quite similar excitation spectra at room temperature. Are there some differences in the excitation spectra of the two emission bands at relatively high temperature (such as 100°C)?

Author Reply: For sample $\text{Li}_2\text{Zn}_{0.85}\text{SiO}_4:0.15\text{Mn}^{2+}$, the emission spectra of the two emission bands at different temperatures ($25\text{--}150^\circ\text{C}$) have been measured and shown in Supplementary Figure 8. No obvious differences can be observed in two excitation bands as the temperature increasing from 25 to 150°C .

Supplementary Figure 8 Excitation spectra of green (530 nm) and red emissions (650 nm) in $\text{Li}_2\text{Zn}_{0.85}\text{SiO}_4:0.15\text{Mn}^{2+}$ at different temperatures (a: 25 °C; b: 50 °C; c: 100 °C; d: 150 °C).

6. The authors should explain more clearly the reason why the emission bands of Mn^{2+} - Mn^{2+} dimer and isolated Mn^{2+} have quite similar excitation spectra in this system.

Author Reply: As provided in the manuscript, Fig. 3c shows a luminescence diagram of the Mn^{2+} - Mn^{2+} dimer. For the Mn^{2+} - Mn^{2+} dimer, the ground state (${}^6\text{A}_1({}^6\text{S}){}^6\text{A}_1({}^6\text{S})$) is formed by the exchange coupling of the ground states (${}^6\text{A}_1({}^6\text{S})$) of Mn1 and Mn2, while the emitting state (${}^6\text{A}_1({}^6\text{S}){}^4\text{T}_1({}^4\text{G})$) resulted from the coupling of the first excited state of Mn1 (${}^4\text{T}_1({}^4\text{G})$) and the ground state of Mn2 (${}^6\text{A}_1({}^6\text{S})$). Under these conditions, the excitation energies for both isolated Mn^{2+} and Mn^{2+} - Mn^{2+} dimer originated from the absorption of Mn1; thus, the green and red emissions possess identical excitation spectra.

7. The authors addressed that Mn^{2+} could occupy both the Li and Zn sites for samples with high concentration of Mn^{2+} . Are there any differences in PL spectra of these two sites?

Author Reply: For the samples $\text{Li}_2\text{Zn}_{1-x}\text{SiO}_4:x\text{Mn}^{2+}$ ($x = 0.01-0.30$), Mn^{2+} will occupy only Zn^{2+} site when the concentration of Mn^{2+} is low ($x < 0.7$). However, as the concentration of Mn^{2+} rises to $x \geq 0.07$, it will simultaneously occupy both Zn^{2+} and Li2 sites and the $\text{Mn}^{2+}(\text{Zn}^{2+})$ - $\text{Mn}^{2+}(\text{Li}2)$ dimer is formed. Therefore, for these samples with relatively high Mn^{2+} concentrations ($x \geq 0.07$), an extra red emission band at ~ 650 nm appeared in the emission spectra in addition to the green emission

at ~530 nm. The green emission at ~530 nm is ascribed to the four coordinated $\text{Mn}^{2+}(\text{Zn}^{2+})$, whereas the red emission (at ~650 nm) cannot be assigned to the Mn^{2+} ion at three or four coordinated Li site. Because the excitation spectrum of the red emission (~650 nm) is quite similar to that of the green emission (~530 nm) (see Figure 3b). Based on the DFT calculations, electron diffraction and XAFS analysis as well as the static and dynamic photoluminescence measurements, the red emission is assigned to the $\text{Mn}^{2+}(\text{Zn}^{2+})\text{-Mn}^{2+}(\text{Li}_2)$ dimer. Although we have not observed the emission of from the isolated Mn^{2+} at Li2 site, the Li-site related red emission 650 nm (from the $\text{Mn}^{2+}\text{-Mn}^{2+}$ dimer) is obviously different to that of the Mn^{2+} at four coordinated Zn^{2+} (~530 nm).

8. For the dimer's emission, does it exist in all Mn^{2+} doped phosphors as the doping level of Mn^{2+} is high enough?

Author Reply: No, it doesn't. The $\text{Mn}^{2+}\text{-Mn}^{2+}$ dimer can be only formed in some special systems when the concentration of Mn^{2+} is heavy enough. The emission center, super-exchange coupled $\text{Mn}^{2+}\text{-Mn}^{2+}$ dimer, might be formed only when two Mn^{2+} ions are close enough (~5 Å) and share anions (A. P. Vink, et al., *J. Electrochem. Soc.* 2001, 148, E313.). If all Mn^{2+} ions are isolated from each other in the compound, the $\text{Mn}^{2+}\text{-Mn}^{2+}$ dimer cannot be formed, even in Mn based compound, such as Cs_3MnBr_5 (B. B. Su et al., *J. Mater. Chem. C*, 2019, 7, 11220.) and the Mn^{2+} based organic-inorganic compounds (V. Morad, et al., *Chem. Mater.* 2019, 31,10161.), etc. Meanwhile, the emission properties of the $\text{Mn}^{2+}\text{-Mn}^{2+}$ dimer is generally affected by the temperature, strength of the super-exchange interactions, as well as the number of $\text{Mn}^{2+}\text{-Mn}^{2+}$ dimers (E. H. Song, et al., *Adv. Sci.* 2015, 2, 1500089.).

9. Please comment on the safety of the proposed optical temperature sensor used in human health monitoring field.

Author Reply: Thanks for your suggestion. The components of the proposed optical temperature sensor, optical encapsulant (OE), polydimethylsiloxane (PDMS) and $\text{Li}_2\text{ZnSiO}_4\text{:Mn}^{2+}$, were fabricated without using any toxic elements. The out cladding (transparent organic polymer) of the fibre is biocompatible and has been frequently employed as the substrate for stretchable electronics (D. H. Kim, et al., *Adv. Mater.* 2008, 20, 4887.; H. Liang, et al., *Adv. Intell. Syst.* 2021, 3, 2100035.). Furthermore, the cytotoxicity experiments also show the fabricated fiber is safe

(Supplementary Figure 12). Therefore, it is safe to use the proposed optical temperature sensor in human health monitoring field. The cytotoxicity experiment of the fibre has been added in revised manuscript and supporting information part (supplementary Figure 12).

Supplementary Figure 12 Cell viability as a function of the time in a culture medium (contained the fabricated fibre).

Peer review comments, second review

Reviewer #1 (Remarks to the Author):

After the revision, this paper is appreciably improved, and, in my opinion, it can be published in the present state.

Reviewer #2 (Remarks to the Author):

The Authors have taken into account my comments in perfect way. The manuscript should be accepted as soon as possible.

Reviewer #3 (Remarks to the Author):

The authors carefully revised the manuscript and cleared the questions/comments raised by the referees. I have no additional comments on it, and suggest that it could be accepted for publication in the current form.

Response to the comments on Nature Communications

Manuscript ID: NCOMMS-21-40097A

Reviewer #1 (Remarks to the Author):

After the revision, this paper is appreciably improved, and, in my opinion, it can be published in the present state.

Author reply: Thank you very much for your positive comments.

Reviewer #2 (Remarks to the Author):

The Authors have taken into account my comments in perfect way. The manuscript should be accepted as soon as possible.

Author reply: Thank you very much for your positive comments.

Reviewer #3 (Remarks to the Author):

The authors carefully revised the manuscript and cleared the questions/comments raised by the referees. I have no additional comments on it, and suggest that it could be accepted for publication in the current form.

Author reply: Thank you very much for your positive comments.